DOI: 10.1038/s41467-018-04291-9　　**OPEN**

Corrected: Author correction

# Ultrafast probes of electron–hole transitions between two atomic layers

Xiewen Wen[1,2], Hailong Chen [3], Tianmin Wu[4], Zhihao Yu [1], Qirong Yang[1], Jingwen Deng[1], Zhengtang Liu[1], Xin Guo[1], Jianxin Guan[1], Xiang Zhang[2], Yongji Gong[2], Jiangtan Yuan[2], Zhuhua Zhang[2], Chongyue Yi[5], Xuefeng Guo [1], Pulickel M. Ajayan[2], Wei Zhuang[6], Zhirong Liu[1], Jun Lou[2] & Junrong Zheng[1]

Phase transitions of electron–hole pairs on semiconductor/conductor interfaces determine fundamental properties of optoelectronics. To investigate interfacial dynamical transitions of charged quasiparticles, however, remains a grand challenge. By employing ultrafast mid-infrared microspectroscopic probes to detect excitonic internal quantum transitions and two-dimensional atomic device fabrications, we are able to directly monitor the interplay between free carriers and insulating interlayer excitons between two atomic layers. Our observations reveal unexpected ultrafast formation of tightly bound interlayer excitons between conducting graphene and semiconducting $MoSe_2$. The result suggests carriers in the doped graphene are no longer massless, and an effective mass as small as one percent of free electron mass is sufficient to confine carriers within a 2D hetero space with energy 10 times larger than the room-temperature thermal energy. The interlayer excitons arise within 1 ps. Their formation effectively blocks charge recombination and improves charge separation efficiency for more than one order of magnitude.

[1] College of Chemistry and Molecular Engineering, Beijing National Laboratory for Molecular Sciences, Peking University, Beijing 100871, China. [2] Department of Materials Science and NanoEngineering, Rice University, 6100 Main Street, Houston, TX 77005-1892, USA. [3] Beijing National Laboratory for Condensed Matter Physics, CAS Key Laboratory of Soft Matter Physics, Institute of Physics, Chinese Academy of Sciences, Beijing 100190, China. [4] Department of Chemical Physics, University of Science and Technology of China, Hefei, Anhui 230026, China. [5] Department of Chemistry, Rice University, 6100 Main Street, Houston, TX 77005-1892, USA. [6] State Key Laboratory of Structural Chemistry, Fujian Institute of Research on the Structure of Matter, Chinese Academy of Sciences, Fuzhou, Fujian 350002, China. These authors contributed equally: Xiewen Wen, Hailong Chen, Tianmin Wu, Zhihao Yu. Correspondence and requests for materials should be addressed to W.Z. (email: wzhuang@fjirsm.ac.cn) or to Z.L. (email: liuzhirong@pku.edu.cn) or to J.L. (email: jlou@rice.edu) or to J.Z. (email: junrong@pku.edu.cn)

In optoelectronic and optoelectro-chemical devices composed of semiconductors and conductors, photo excitation can generate transient conducting free carriers by promoting electrons from the occupied valence band into the empty conduction band of the semiconductors[1]. The electrostatic interaction between the depleted valence band states (holes) and excited electrons can lead to the formation of long-lived insulating excitons[2–4]. The evolution of these electron/hole states on interfaces in the devices, which arises from interactions among phonons, photons, and charged quasiparticles, determine their fundamental properties, e.g., luminescence, heat generation, charge separation and transport, redox reaction kinetics, and outcomes[5–7]. In the past, the interfacial dynamics of excited optoelectronic materials were often studied on bulk or nano-sized samples that contain both interfacial and bulk components with absorption, photoluminescence, and visible pump/probe experiments close to the bandgap[1,8,9]. Despite providing a wealth of important information, these techniques are sensitive to creation and destruction of excitons and thus only probe the quasiparticles with negligible center-of-mass momentum because of the small photon momentum. However, the majority of excitons can assume finite momenta as a result of scattering and is optically dark in these measurements[10,11].

In contrast, through detecting internal quantum transitions[11–14], photons with lower energy, e.g., the near- and mid-infrared (IR) or THz, can probe photo-generated excitons at center-of-mass momenta well outside the optically accessible range (Fig. 1a), allowing precise measurements and direct monitoring of excitonic energy and dynamics without interference from other species. The method can effectively avoid mixing together the signals of free carriers, excitons, and defect-trapped carriers that usually occur in visible pump/probe experiments due to the high detection photon energy[15]. On the other hand, the advent of two-dimensional (2D) heterostructures composed of only two atomic layers[16,17] provides an ideal laboratory for the studies of pure interfacial dynamical transitions of charged quasiparticles. However, limited by spatial resolution, it remains a challenge for regular IR setups to probe electron/hole pairs in 2D heterostructures of which the sizes are typically smaller than the IR beam size. In this work, combining microscope, ultrafast visible/near-infrared/mid-infrared frequency-mixed spectroscopy (Fig. 1b) and state-of-the-art 2D atomic device fabrications, we are able to conduct the first experiments directly monitoring the phase transitions of electron/hole gas between two atomic layers. Studies under various conditions reveal unexpected ultrafast transformation of conducting free carriers into tightly bound insulating interlayer excitons across the conducting graphene and semiconducting $MoSe_2$ interface.

## Results

**Photoluminescence of $MoSe_2$ is quenched by graphene.** In our experiments, photo-excited electrons/holes are quantum confined within the 2D space of two atomic layers. We investigate heterostructures composed of one monolayer of $MoSe_2$ and another monolayer of p-doped graphene (Fig. 2a). The monolayer $MoSe_2$ is a direct bandgap semiconductor[2,18]. It has a very small excitonic Bohr radius and anomalously strong interband optical absorptions[2]. Its excitonic binding energy is ~0.6 eV (Supplementary Fig. 24), orders of magnitude larger than bulk materials[12,19,20]. The large binding energy allows the intralayer excitons of $MoSe_2$ to remain stable at room temperature. The monolayer graphene is a semi-metallic conductor[21] in which optical excitations generate free carriers that are rapidly thermalized to reach a new Fermi–Dirac electronic distribution[22]. Because of its gapless band structure, the conducting free carriers in graphene do not transform into insulating excitons[23–25]. The distinctly different opto-electro properties of graphene and $MoSe_2$ and other direct bandgap 2D transition metal dichalcogenide (TMDC) materials[2,18] promise ideal combinations for atomically thin optoelectronic devices in which 2D TMDC functions as photon absorbers and graphene layers serve as transparent electrodes[16,17].

The band alignment of $MoSe_2$/graphene heterostructure is illustrated in Fig. 2b. The separation of conduction band

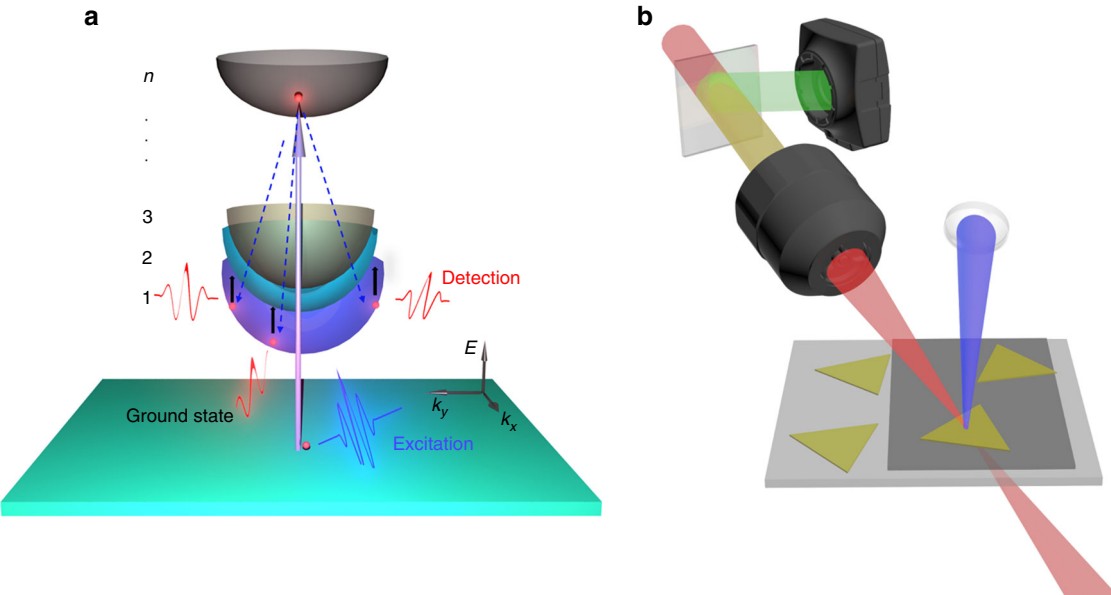

**Fig. 1** Illustration of experimental setup and principle. **a** Mid-IR probes the excitonic internal quantum transition 1s-2p with a broad momentum distribution. Optical excitation generates free carriers with negligible momentum change. Scattering of free carriers (dash lines) into the exciton states involves large momentum transfers. **b** Illustration of the experimental setup. The IR probe is focused by an objective lens and transmits through the sample, detected by a MCT array detector behind the sample. The reflected visible light through the objective lens is collected by CCD to position the light/matter interaction spot. By tuning the sample stage, heterostructure, graphene, and $MoSe_2$ can be respectively studied on the same $CaF_2$ substrate. Triangles and the light-black layer represent single-layer $MoSe_2$ and graphene, respectively

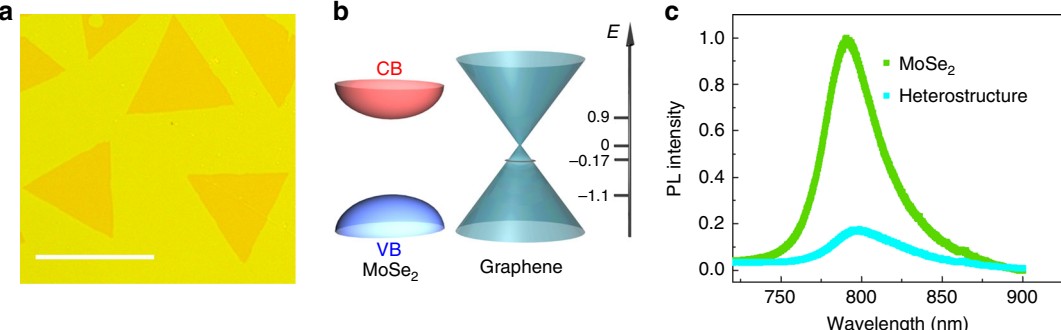

**Fig. 2** MoSe2/graphene heterostructures. **a** Optical image of the MoSe2/graphene heterostructures on a CaF2 window. The triangles are MoSe2 below a continuous monolayer of graphene. Scale bar represents 100 μm. **b** Band alignment of the heterostructure. The energy levels CBM (conduction band minimum of MoSe2), Dirac point and Fermi level of graphene, and VBM (valence band maximum of MoSe2) in eV are listed in the order from top to bottom on the right. **c** Photoluminescence (PL) spectra of the MoSe2 monolayer and MoSe2/graphene heterostructure. PL is severely quenched in the heterostructure

minimum (CBM) and valence band maximum (VBM) of MoSe2 is around 2.0 eV[19,20,26]. The Dirac point of graphene is 0.9 eV below CBM of MoSe2. The Fermi level of graphene in heterostructure is -0.17 eV, 0.02 eV higher than the monolayer graphene because of charge transfer from the n-doped MoSe2. The photoluminescence (Fig. 2c) of the heterostructure at 795 nm (1.56 eV) is dominated by the excitonic 1s transition of MoSe2. It is severely quenched by the presence of graphene because of efficient interlayer charge and energy transfers[27].

**Excitation below MoSe2 transition energy**. In ultrafast experiments (Fig. 1b), the sample is excited with 40-femtosecond near infrared and blue pulses at 1200 nm (1.03 eV, below the MoSe2 bandgap) and 400 nm (3.1 eV), respectively. The optical density change (proportional to the optical conductivity change, Supplementary Equation 5) following excitation is monitored with a mid-IR pulse. Figure 3a, b display optical density changes of a monolayer graphene and a graphene/MoSe2 heterostructure detected in the mid IR range 1950–2230 cm$^{-1}$ following excitation at 1.03 eV. Transient spectra and kinetic data are provided in SI (Supplementary Fig. 1). To illustrate the changes of detection frequency dependence in the samples, the maximum intensity at each waiting time is set to be 1. The spectrum of monolayer graphene remains flat across the entire detection frequency range and independent of waiting time (Fig. 3a). In contrast, the spectrum of the heterostructure is time dependent. With the increase of delay time, the relative intensity in low frequencies drops significantly (Fig. 3b). The spectra at 16 ps (Fig. 3c) reveal the striking difference between heterostructure and graphene: a resonant peak centered at 2185 cm$^{-1}$ (0.27 eV) with a Lorentzian width of 280 cm$^{-1}$ remains in heterostructure that lasts for more than 20 ps (Fig. 3c) while the graphene signal is already zero. The huge signal difference can be further revealed by the detection frequency dependent dynamics. Detected at 2185 cm$^{-1}$ (Fig. 3d), the heterostructure dynamics is apparently slower than graphene and with a nonzero tail (Fig. 3e). However, at a lower detection frequency, 1860 cm$^{-1}$ (Fig. 3f), the nonzero tail of the heterostructure signal disappears, and its dynamics becomes faster. In fact, at this detection frequency, the heterostructure and graphene follow essentially the same dynamics, and both are slower than the intralayer free-carrier dynamics[4] in monolayer MoSe2 excited with 3.1 eV photons. No signal is observed for MoSe2 monolayer with the 1.03 eV excitation. The signal of monolayer MoSe2 excited with 3.1 eV photons rises apparently slower than those of samples excited with 1.03 eV photons. The difference mainly originates from different signal mechanisms. The signal with the 3.1 eV excitation directly comes from the absorptions of both fast

generated free carriers and excitons generated in a subsequent slower process, but the signal with 1.03 eV excitation is from the very fast electronic thermal redistribution in graphene as discussed in the following.

**Interlayer excitonic internal 1s-2p quantum transition**. The huge difference in signal reveals distinct evolutions of electrons/holes in graphene and heterostructure. The excitation photon energy 1.03 eV is lower than the excitonic transition energy (1.59 eV) of MoSe2, and thus negligible electron/hole pairs are generated in the monolayer MoSe2. In the monolayer graphene, the excitation generates $5.2 \times 10^{12}$ electrons-holes/cm$^2$ with energy separation equally above and below the Dirac point. Energy rapidly thermalizes among the charge carriers and relaxes through electron/phonon couplings with the transition matrix element[28,29]

$$M_{\mathbf{k'},\mathbf{k}}^{(TO\&LO)} \approx 3\eta \sqrt{\frac{\hbar}{4 M_C \omega_{phonon}}} \quad (1)$$

where $\eta$ is the electron–phonon coupling parameter, $\omega_{phonon}$ is the phonon angular frequency, and $M_C$ is the mass of a carbon atom. The processes cause an electronic temperature change and a Fermi–Dirac electronic population redistribution, leading to the optical conductivity change that is determined by both intraband and interband transitions[30]:

$$\sigma_{inter}(\omega) = i \frac{e^2 \hbar \omega}{\pi \hbar} \int_0^{+\infty} d\varepsilon \frac{1}{(2\varepsilon)^2 - (\hbar\omega + i\Gamma)^2} [f_{FD}(\varepsilon - \mu) - f_{FD}(-\varepsilon - \mu)]$$

$$\sigma_{intra}(\omega) = i \frac{e^2/\pi\hbar}{\hbar\omega + i\hbar/\tau_e} \int_0^{+\infty} d\varepsilon [f_{FD}(\varepsilon - \mu) + 1 - f_{FD}(-\varepsilon - \mu)]$$

$$(2)$$

where $f_{FD}$ is the Fermi–Dirac distribution function, $\mu$ is the chemical potential (Fermi energy), and $e$ is the elementary charge, $\Gamma$ is the broadening of the interband transitions, whereas $\tau_e$ is the relaxation time due to intraband carrier scattering. Quantitative analyses of the graphene signal according to Eqs. 1 and 2 describe the experimental results very well (Fig. 3d), yielding a Fermi level of $-0.19$ eV, the electron/hole population dynamics, and the temperature dynamics of graphene. Details of the analyses are provided in SI. The decay of free electrons/holes in graphene is extremely fast, with a time constant of 120 fs (Fig. 3h, i, navy). These obtained graphene properties are similar to the previously reported results[22,31].

In the heterostructure, the detection frequency dependent signals and dynamics (Fig. 3b–f) resemble those free carrier/exciton transitions of GaAs quantum wells observed in the THz

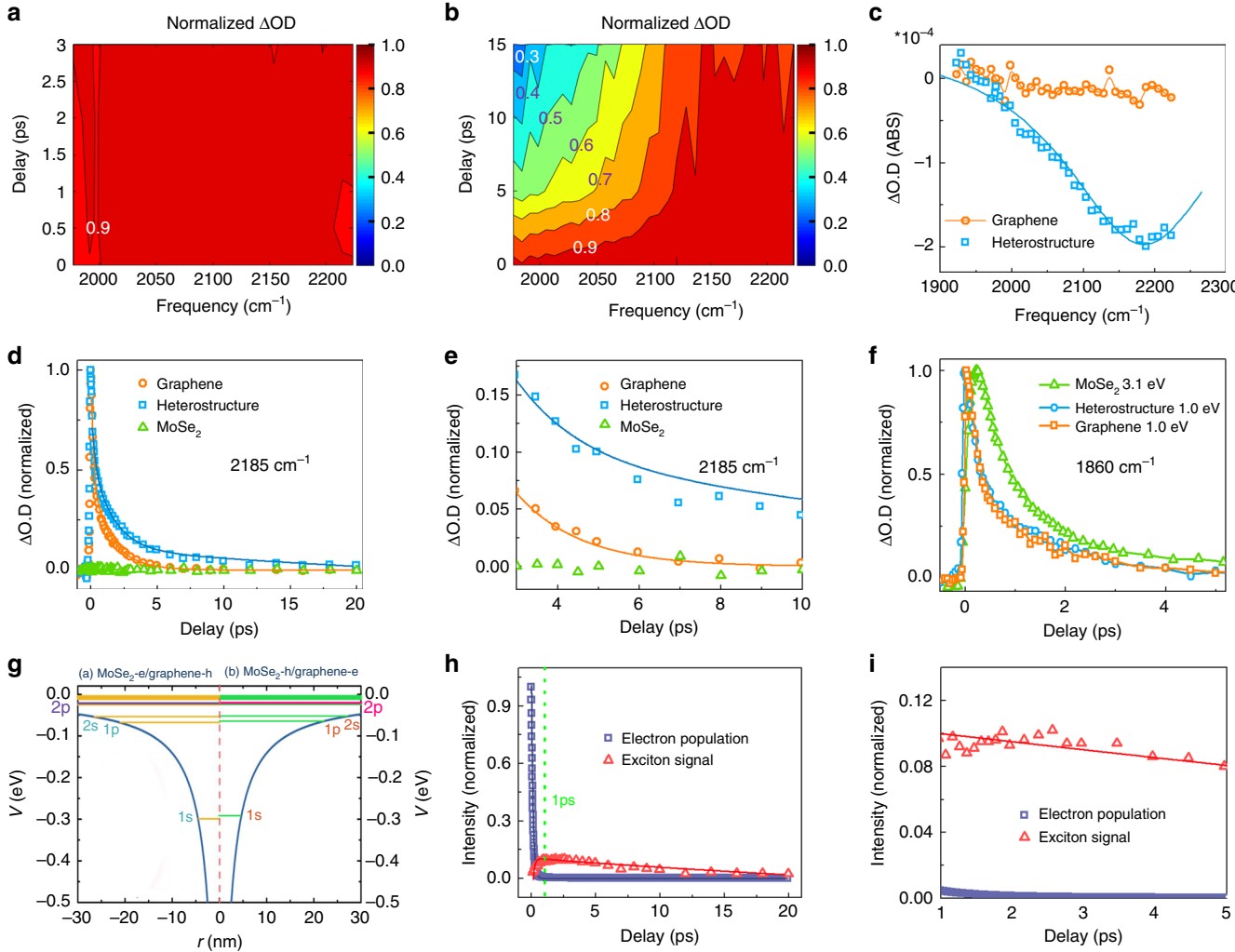

**Fig. 3** Ultrafast measurements reveal the formation of interlayer excitons. Waiting time-dependent spectra of **a** graphene monolayer and **b** MoSe$_2$/graphene heterostructure excited by 1.03 eV photons. The maximum intensity at each waiting time is set to be 1. Thus, the figures only reflect the changes of detection frequency dependence rather than decay dynamics. Each contour represents 10% intensity change. **c** Spectra of MoSe$_2$/graphene heterostructure and graphene monolayer at 16 ps after excitation with 1.03 eV photons. The graphene signal (both dots and curve) is already zero, but the heterostructure signal is at peak. The peak is fit with a Lorentzian centered at 2185 cm$^{-1}$ with a width 280 cm$^{-1}$. Dots are data and the curve is calculation. **d** Waiting time-dependent transient IR signals detected at 2185 cm$^{-1}$ of monolayer graphene, MoSe$_2$/graphene heterostructure, and MoSe$_2$ monolayer with 1.03 eV excitation. The dynamics of the heterostructure is apparently slower than graphene. Dots are experimental data, and lines are theoretical calculations. The negligibly small signal of MoSe$_2$ is normalized to the maximum intensity of the graphene signal. **e** Enlarged waiting time-dependent transient IR signals after 3 ps in Fig. 3d, which illustrates the nonzero tail of the heterostructure signal. **f** Waiting time-dependent transient IR signals detected at 1860 cm$^{-1}$ of the monolayer graphene and MoSe$_2$/graphene heterostructure with 1.03 eV excitation, and MoSe$_2$ monolayer with 3.1 eV excitation. Different from the 2185 cm$^{-1}$ detection, within experimental uncertainty the dynamics of graphene and heterostructure are the same. Both are slower than the free-carrier dynamics in MoSe$_2$ monolayer with 3.1 eV excitation. **g** Calculated interlayer exciton energy levels of MoSe$_2$/graphene heterostructure with graphene's Fermi level at -0.17 eV with a 2D model. The calculations show a binding energy of about 0.3 eV for the interlayer excitons. **h** Calculated waiting time dependence of electronic dynamics (navy) for graphene with Fermi level at −0.17 eV, and interlayer exciton signal (red). Lines are kinetic analyses. **i** Calculated huge population difference between excitons and free carrier after 1 ps in Fig. 3h, when the concentration of free carriers is close to 0

range, where the resonant peak is assigned to the excitonic internal quantum 1s-2p transition[12,32]. A similar 1s-2p transition peak was also observed in the mid-IR range for monolayer WSe$_2$ on a diamond[11]. In this work, different from the two previous experiments, the graphene/MoSe$_2$ heterostructure sample is not a pure substance, but a combination of two monolayers. Thus, the nature of the exciton that give rise to the resonant peak at 2185 cm$^{-1}$ can have several possible origins: 1, the internal neutral and charged excitons, e.g., regular excitons and trions[33] in graphene; 2, the internal neutral and charged excitons in MoSe$_2$; and 3, the interlayer neutral and charged excitons.

In the following, combining experimental data and literature, neutral interlayer excitons are concluded to be the dominant species that produce the resonant signals. First of all, the contribution of the charged excitons (including trions and other excitons composed of more than three carriers) is very minor in room temperature optical/IR pump/probe experiments. This has been completely addressed in the SI of the previous work[11]. In brief, the binding energy of trions is only about 30 meV[33]. The binding energy of excitons with more particles is even smaller. At room temperature, where our experiments were carried out, most of the trions had dissociated due to thermal motions[11,34], which

has been verified with temperature-dependent photoluminescence[11,34]. Second, the intralayer exciton in graphene is not likely. Although the carriers in doped graphene are no longer massless due to the Fermi level shift from the Dirac point and a slight symmetry breaking[35], tightly bound excitons cannot form because the bandgap of graphene in the heterostructure is too small (4.4 meV, Supplementary Fig. 25), compared to room temperature thermal energy (26 meV)[35]. Third, intralayer excitons in $MoSe_2$ (with interlayer effects that cause its binding energy to drop) are not likely either. The combination of the experiments with 3.1 and 1.03 eV excitations in Fig. 3d, f rule out this possibility. 1.03 eV is significantly lower than the bandgap of $MoSe_2$. Excitation with it cannot produce detectible amounts of free carriers or excitons in $MoSe_2$ (green dots in Fig. 3d). In the heterostructure, if the peak at $2185\ cm^{-1}$ (Fig. 3c) was due to the intralayer excitons in $MoSe_2$, the excitons would come from the holes and electrons transferred from graphene. After the carriers transfer to $MoSe_2$, they would behave like free carriers[4], and then these free carriers would decay to form excitons and some possible trapped states inside $MoSe_2$. In the process, the free-carrier dynamics inside $MoSe_2$ would be similar to, or slightly slower (due to lower energy)[4] than the free carriers created with 3.1 eV excitation in the monolayer $MoSe_2$. However, experimentally, the free-carrier dynamics detected at $1860\ cm^{-1}$ of the heterostructure excited with 1.03 eV is significantly faster than that inside the monolayer $MoSe_2$, and essentially the same as graphene (Fig. 3f). The detection energy $1860\ cm^{-1}$ is lower than the excitonic internal quantum transition energy, $2185\ cm^{-1}$. Therefore, signals at $1860\ cm^{-1}$ cannot probe excitons, but only reflect free-carrier dynamics. This comparison of free-carrier dynamics rules out the possibility that the observed dynamics of the heterostructure is mainly from $MoSe_2$ intralayer dynamics, indicating that the resonant peak at $2185\ cm^{-1}$ should not belong to $MoSe_2$ intralayer excitons. In fact, the intralayer excitonic internal quantum transition of monolayer $MoSe_2$ appears at a much higher energy, 0.55 eV (Supplementary Fig. 24), further supporting that the resonant peak at $2185\ cm^{-1}$ observed in the heterostructure should not belong to the $MoSe_2$ intralayer excitons. In conclusion, all these systematic experimental results suggest that the neutral interlayer excitons are the dominant species leading to the resonant peak at $2185\ cm^{-1}$. The similarity of the free-carrier dynamics between heterostructure and graphene (Fig. 3f) is interesting and worth further discussion. A similar phenomenon in a $WS_2/MoS_2$ heterostructure was also observed in our previous work[4] in the formation process of interlayer excitons, the free-carrier dynamics inside the heterostructure follow the faster dynamics ($MoS_2$) of the two monolayers. It is conceivable that the formation of interlayer excitons need carriers to diffuse in both the layers and the faster partner dominates.

However, the interlayer excitons involving graphene with binding energy larger than $2185\ cm^{-1}$ (0.27 eV) seem to be unlikely and completely unexpected if estimated with conventional hydrogen-like 3D model. It is well known that pristine graphene has massless carriers, which cannot form excitons[36]. In most CVD grown graphene, which is effectively doped, like the one used in our experiments, the Fermi level is not at the Dirac point. Because of the Fermi level shift and the slight symmetry breaking due to the formation of heterostructure[35], the carriers are no longer massless (Supplementary Fig. 25). The effective mass of the carriers in graphene with Fermi level −0.17 eV is between 0.01–0.03 $m_0$ ($m_0$, the free electron mass), similar to 0.012 $m_0$ reported for an epitaxy-grown graphene[37]. Estimated with this effective mass and the conventional 3D hydrogen-like model in which the binding energy is linearly proportional to the reduced mass, the binding energy of graphene/$MoSe_2$ would be

only 1/60 of the $MoSe_2$ intralayer excitons. The binding energy of $MoSe_2$ intralayer excitons has been computed to be between 0.4 and 0.6 eV[19]. Our experiments (Supplementary Fig. 24) also verify that it is ~0.60 eV, under our experimental conditions. This would suggest that the estimated binding energy of the interlayer excitons should be smaller than 0.01 eV, significantly smaller than 0.27 eV, the observed excitonic internal quantum transition energy. The key to solve the contradiction lies in the fact that the atomic layer 2D excitons cannot be described by 3D hydrogen-like models[38,39]. Instead, the 2D models must be applied[38,39]. In the 2D models, the binding energy is no longer linearly proportional to the effective mass[38,39]. Estimated from a 2D model, the interlayer excitonic binding energy is much larger, about 1/2–1/4 of the $MoSe_2$ intralayer excitonic binding energy, which is around 0.1–0.3 eV. This estimated value is close to the excitonic internal quantum transition energy 0.27 eV observed. Using a 2D model that has been previously tested on other 2D materials[4,38], the binding energy of $MoSe_2$/graphene interlayer excitons is calculated to be 0.3 eV (Fig. 3g). The calculations match the experimental results very well. Calculation details are in SI (Supplementary Figs 8–23).

In summary, the excitation of $MoSe_2$/graphene heterostructure with 1.03 eV photons results in the formation of interlayer excitons. The dynamical process can be visualized in terms of free-carrier/exciton transitions. The photon excitation creates free carriers in graphene. The semi-instantaneous thermalization of carriers[22] leads to a new Fermi–Dirac distribution, in which the number of holes at energy below the VBM of $MoSe_2$ is around four times more than that of electrons above the CBM of $MoSe_2$. These charged carriers rapidly transfer between the two layers[4,27] and reach quasi-equilibrium, finally resulting in holes dominantly in $MoSe_2$ and electrons in graphene. Oppositely charged quasiparticles on the two layers attract each other and form interlayer excitons. The excitonic 1s-2p transition leads to the appearance of a resonant peak centered at $2185\ cm^{-1}$ (0.27 eV) (Fig. 3c). As shown in Fig. 3h (navy line vs red curve), the interlayer exciton's lifetime is much longer than the free carriers, similar to that of $WS_2/MoS_2$ heterostructure[4]. At time zero, most electrons/holes in the heterostructure are free carriers, while within 1 ps the majority of the remaining electron/hole pairs are already interlayer excitons (Fig. 3i). Kinetic analyses of the experimental results normalized with optical cross sections show that the interlayer exciton has a formation time constant 0.49 ± 0.27 ps, with a lifetime 65 ± 20 ps (red in Fig. 3h, dots-experiments and line-theory). The results indicate that around 25% (120 fs/490 fs) of the free carriers' photogenerated in graphene transform into interlayer excitons within hundreds of fs, and by the time the majority of other free carriers have recombined and released heat in graphene. Calculation details and more discussions are provided in SI (Supplementary Figs 1–7). The surprising charge flow suggested by ultrafast measurements is consistent with photocurrent measurements, in which the photocurrent is measured between graphene on the top of $MoSe_2$ and graphene on the top of Si. As displayed in Supplementary Fig. 27 (with a thin layer of $SiO_2$), the 980 nm photoexcitation, which excites only graphene but not $MoSe_2$, produces external photocurrent flowing from the portion of graphene that is on the top of Si (with a thin layer of $SiO_2$) to that on the top of $MoSe_2$, which is a clear signal that the quasi-equilibrium of $MoSe_2$ (hole)/graphene (electron) inside the heterostructure is reached by the 980 nm photoexcitation. The results have a very interesting indication: under the experimental condition, graphene functions as a photon absorber and $MoSe_2$ serves as a charge acceptor, rather than their traditional roles, in which graphene is the transparent electrode and $MoSe_2$ is the chromophore.

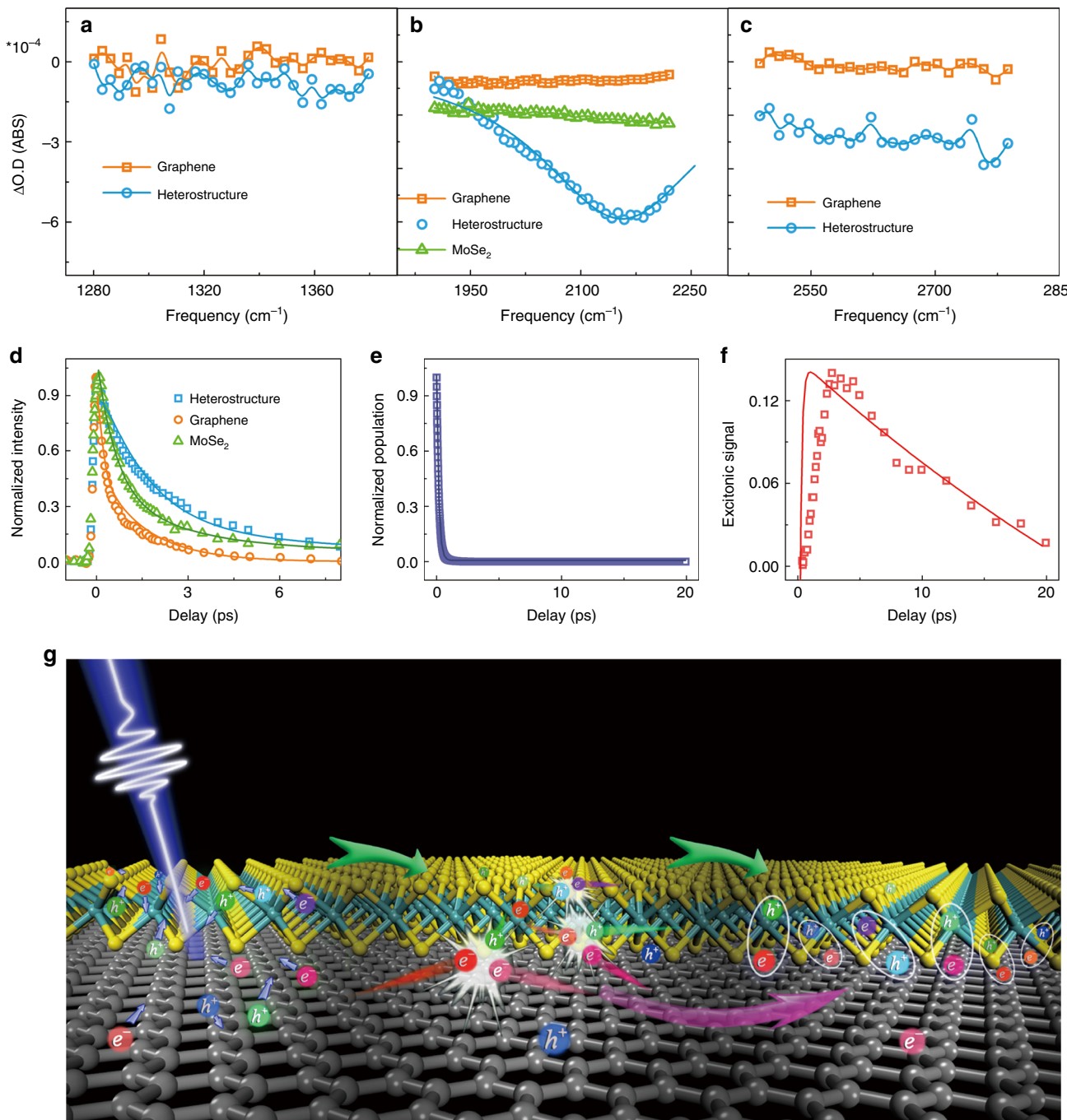

**Fig. 4** Excitation with 3.1 eV also leads to the formation of interlayer excitons. **a** Spectra of MoSe₂/graphene heterostructure and graphene monolayer at 16 ps after excitation with 3.1 eV photons detected in the frequency range 1280–1380 cm⁻¹ below the excitonic 1s-2p transition frequency. Both signals are zero. **b** Spectra of MoSe2/graphene heterostructure, MoSe₂, and graphene at 16 ps after excitation with 3.1 eV photons detected in the frequency range 1900–2230 cm⁻¹ covering the excitonic 1s-2p transition frequency. Both graphene and MoSe₂ spectra are flat, whereas that of heterostructure is a peak with a much higher intensity centered at 2156 cm⁻¹, with a Lorentzian width 278 cm⁻¹. **c** Spectra of MoSe₂/graphene heterostructure and graphene monolayer at 16 ps after excitation with 3.1 eV photons detected in the frequency range 2450–2800 cm⁻¹, above the excitonic 1s-2p transition frequency. The graphene signal is zero, and that of the heterostructure is a nonzero line because of the transition to higher bound and unbound states12. **d** Waiting time-dependent normalized transient IR signals detected at 2185 cm⁻¹ of MoSe₂/graphene heterostructure, MoSe₂, and graphene. The dynamics of heterostructure is the slowest. The initial absolute intensity ratio of the three samples is 3.4/1.8/1=heterostructure/graphene/MoSe₂. Dots are data, and lines are theoretical calculations. **e** The electronic dynamics in graphene of heterostructure. Dots are calculations and the line is fitting. **f** The interlayer excitonic signal in the heterostructure. Dots are experimental data and the line is kinetic calculation. **g** Illustration of electron/hole gas transition in the heterostructure. An electron/hole pair in ellipse represents an exciton. Excitation with photo energy (3.1 eV) higher than MoSe2 bandgap creates free carriers in both MoSe₂ and graphene. The carriers transfer between the two layers. The carriers collide with each other and transfer energy and momenta so that phonon motions are not necessary for the ultrafast formation of interlayer excitons. Because of the band alignment, more electrons are on the graphene side and more holes are on the MoSe₂ side

**Excitation above MoSe₂ transition energy**. The ultrafast formation of interlayer excitons is also observed when the majority of free carriers is from $MoSe_2$. Figure 4a–c displays the spectra of graphene, $MoSe_2$ and $MoSe_2$/graphene heterostructure detected in three different IR ranges at 16 ps after excitation with 3.1 eV photons. In the low-frequency range (1280–1380 $cm^{-1}$), signals of both graphene and heterostructure are zero. Between 1900 and 2230 $cm^{-1}$, the spectrum of graphene is very close to zero, whereas the spectrum of the heterostructure is at peak. In the high frequency range (2500–2800 $cm^{-1}$), the signal of graphene remains zero, but that of heterostructure is a flat spectrum with an amplitude of about 30–40% of the peak value at 2156 $cm^{-1}$. Similar to those following 1.03 eV excitation (Fig. 3c), the peak centered at 2156 $cm^{-1}$ with a width 278 $cm^{-1}$ is attributed to the interlayer excitonic transition[12,32]. The central frequency slightly redshifts from 2185 $cm^{-1}$ with 1.03 eV excitation. This frequency difference is probably because of the experimental uncertainty, rather than excitonic difference. Our frequency resolution is about 9 $cm^{-1}$, and the broadband super continuum probe pulse has a spatial dependence of frequency distribution that can lead to the signal intensity at a certain frequency, dependent on the focus condition. These two factors together can cause the peak frequency to shift for 10% of its width. Therefore, within the experimental uncertainty, we would conclude that the peak frequency and width are the same as those excited with 1.03 eV photons. The flat spectrum at higher frequencies corresponds to transitions into higher-energy bound and continuum states[12,32]. The insulating nature of the interlayer exciton is apparent from the vanishing optical conductivity at low frequencies (Fig. 4a).

3.1 eV is higher than the bandgap of $MoSe_2$. In the heterostructure, the photo excitations not only generate free carriers in graphene, but also promote electrons into the unbound continuum of $MoSe_2$. The excited quasiparticles exchange between the two monolayers, resulting in electrons dominantly in graphene and holes mainly in $MoSe_2$ which attract each other across layers. Such a charge separation produces a photocurrent, in which the electrons flow from graphene to $MoSe_2$ in the external circuit. The photocurrent results are shown in Supplementary Fig. 28.

Because the absorption of $MoSe_2$ (14%) is significantly higher than that of graphene (4.14%) at 3.1 eV[40–42], overall the interlayer charge transfers result in higher electronic temperature in graphene. Excluding the free carriers remaining in $MoSe_2$, the effect is around equal to 200% excitation flux increase for graphene monolayer, assuming semi-instantaneous interlayer charge transfers[4,27] and electronic thermalization[22]. Our calculations show that the temperature increase with additional 200% flux slows the graphene electronic dynamics for about 30%, from 130 to 170 fs (Supplementary Fig. 5). The temperature effect together with the formation of long-lived interlayer excitons causes the dynamics of the heterostructure to be significantly slower than the monolayer graphene (Fig. 4d). The heterostructure dynamics is even slower than that of $MoSe_2$ monolayer, which only reflects the dynamics of the free carriers in $MoSe_2$[4]. Kinetic analyses show that the interlayer exciton formation time constant is $0.51 \pm 0.28$ ps, with an excitonic lifetime $55 \pm 20$ ps (red in Fig. 4e, dots-experiments and line-theory.), indicating around 33% of the free carriers photogenerated form the interlayer excitons. The exciton dynamics are similar to excitation with 1.03 eV photons. However, because the free-carrier dynamics is slower with 3.1 eV excitation, more excitons are generated.

## Discussion

The fast formation of $MoSe_2$/graphene interlayer excitons within hundreds of fs is surprising. On one hand, previous theories of exciton formation consider only the extreme dilute limit, where independent charge carriers interact with phonons[43,44], predicting much slower ps to hundreds of ps exciton formation dynamics. On the other hand, the existence of interlayer excitons is not expected in a semiconductor/graphene heterostructure, where the Fermi level of graphene typically lies between VBM and CBM of the semiconductor layer[45], because the aforementioned carriers in graphene is nearly massless, and the ultrafast (<50 fs) interlayer charge transfers[4,27] send both electrons and holes into graphene[45], where they recombine within a couple of hundred fs[23,25]. The commonly accepted but oversimplified pictures, however, cannot account for the complex many-body dynamics in real systems and our experiments. Quasiparticles that excitons are composed of cannot be treated separately. In a photo-excited electron/hole gas, the ultrafast Coulomb interactions result in fundamentally new properties through pairing and higher-order correlations[46]. Some of these can be intuitively visualized by processes such as a pair of free electron and hole forming an exciton by transferring their energy and momenta to other free carriers or redistributions among pairs through collisions (Fig. 4g). The many-body Coulomb interactions naturally explain the ultrafast formation of interlayer excitons observed in our experiments. Once electrons and holes form excitons, they cannot recombine directly because of the in-plane momentum mismatch that is due to the random relative orientation between graphene and $MoSe_2$. The carriers need to either compensate for the momentum mismatch or cross the binding barrier to recombine, which takes extra steps and time. Therefore, the formation of interlayer excitons effectively retains the charge separation across the interface of the heterostructure by significantly slowing down the otherwise ultrafast interlayer charge transfers.

The transition of conducting free carriers into insulating interlayer excitons observed in the semiconductor/semimetallic 2D heterostructure reveals the significance of many-body interactions in mediating and thus producing sophisticated electronic dynamics and states on interfaces. The unexpected formation of the tightly bound interlayer excitons suggest that carriers in the doped graphene of the heterostructure is no longer massless, and an effective mass as small as one or two percent of $m_0$ is sufficient to confine carriers within a 2D hetero space with energy 10 times larger than the thermal energy at room temperature. One important consequence of the ultrafast formation of interlayer excitons is that it significantly improves the intrinsic charge separation efficiency on graphene interface. Without interlayer excitons, the photo-generated carriers would transfer to and recombine in graphene within a couple of hundred fs. The long-lived interlayer excitons (>50 ps) keep the charged quasiparticles on different layers for more than 20 times longer. The greatly improved intrinsic charge separation efficiency is important for many applications that utilize the conversion of photo energy into electronic energy or electricity, e.g., solar cells, photo-detections, and photo-electrochemistry. We anticipate that the micro-spectroscopic IR response uncovered here will enable realtime and precise studies of the new non-equilibrium states on atomic interfaces that are critical for future developments and applications of atomically thin and other optoelectronic devices, but cannot be studied with other means.

## Methods

**MoSe₂ monolayer growth**. We follow the CVD method of growing high-quality monolayer $MoSe_2$ in our prior publication[47] using Selenium pellets and Molybdenum oxide as sources.

**Graphene monolayer growth**. CVD method was used to grow graphene on an electropolished copper foil with methane as a precursor. The copper foil was annealed at 1000 °C for 20 min, followed by graphene growth for 9 min using 3.5 sccm of methane under the same temperature. 15% $H_2$/Ar was used as a carrier gas and the pressure was kept around 1 torr during the whole process.

**Heterostructure preparation.** The as-grown $MoSe_2$ are transferred onto the $CaF_2$ substrates for transmission mode laser experiments. The process is: at first, a thin layer of PMMA (PMMA-B4 4%wt) is spin-coated onto the sample on $SiO_2$ substrate twice at the speed of 4000 rpm, and then the sample is carefully placed in a Buffer HF (1:5) and allowed to float. After 20 h, the PMMA sheets are separated from the $SiO_2$ substrate and fished with glass plate, and then moved to Di-water for rinsing for a few times. The pre-cleaned $CaF_2$ windows are used to fish the PMMA sheets. The $CaF_2$ windows with PMMA and samples stay in room temperature and vacuum for 24 h. Finally, acetone is used to remove the PMMA. The $MoSe_2$ triangles on the $CaF_2$ substrate are confirmed by optical microscope. Graphene on the Cu foils spun with PMMA is dissolved in $Cu_2Fe_3$, and the PMMA sheet is fished and transferred like the same method above onto the $MoSe_2$ monolayers. PMMA is removed with acetone. Finally, the sample is annealed in vacuum at 300 °C for 3 h.

**Raman and photoluminescence measurements.** Raman and PL spectroscopy were carried out using a Horiba Jobin Yvon LabRAM HR-Evolution Raman microscope. The excitation light is a 532 nm laser, with an estimated laser spot size of 1 μm and the laser power of 1 mW.

**Ultrafast visible-NIR/infrared microspectroscopy.** The experimental setup[4] of the ultrafast visible-NIR/infrared microspectroscopy is illustrated in Fig. 1b. Briefly, the output of a femtosecond amplifier laser system (at a repetition rate of 1 kHz, 1.6 mJ energy per pulse, 800 nm central wavelength, and a pulse duration of ~40 fs, Uptek Solutions Inc.) is split into two parts. One is used to pump a home-built nonlinear optical parametric amplifier to generate visible and near-IR-1 laser pulses with tunable wavelengths, and the other is directed to generate an ultra-broadband super-continuum pulse, which covers almost the whole mid-IR region[48,49] or a near-IR-2 pulse. In ultrafast experiments, the visible or NIR-1 pulse is the pump light with the central wavelength and excitation power adjusted based on need. The interaction spot on the samples varies from 120 to 250 micron. The mid-IR super-continuum pulse or NIR-2 pulse acts as the probe light, which is focused at the sample by the reflective objective lens (15X/0.28NA, Edmund Optics Inc.) to reduce the spot size to the level of the sample area (<40 μm). A 300-megapixel microscope digital camera is used to align the pump/probe beam to proper sample area. The probe light is detected by a liquid-nitrogen-cooled mercury-cadmium-telluride (MCT) array detector or InSb detector after frequency resolved by a spectrograph with a resolution of $1-3$ $cm^{-1}$, which is dependent on the central frequency. The time delay between the pump light and probe light is controlled by a motorized delay stage.

**Data availability.** The data that support the findings of this study are available from the corresponding author upon request.

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

## Acknowledgements

This material is based upon work supported by the National Science Foundation of China (NSFC-21627805, 21673004, 21773002, 21373201, and 21033008) and MOST (2017YFA0204702) China, the Strategic Priority Research Program of the Chinese Academy of Sciences (XDB20000000 and XDB10040304), the AFOSR grant FA9550-14-1-0268, the Welch Foundation grant C-1716. Discussions from Prof. Feng Wang at UC Berkley are appreciated. We thank Qingye Zhu, Ruiheng Wu, and Haowen Zhou for their help in preparing Figs. 1 and 2.

## Author contributions

J.Z., X.W., and H.C. designed the experiments. J.Z. supervised the project. X.W., H.C., Z.Y., Zhengtang L., X.G., J.D., and J.Z. performed ultrafast experiments. X.W., Y.G., X.Z., J.Y., C.Y., J.L., and P.A. prepared materials. Q.Y., J.G., and X.G. performed photocurrent measurements. J.Z. and Zhirong L. analyzed the data. Z.Z., T.W., and W.Z. performed band structure and binding energy calculations. J.Z., Zhirong L., W.Z., J.L., Y.Z., X.W., and H.C. prepared and revised the manuscript.

## Additional information

**Competing interests:** The authors declare no competing interests.

