## [Peer Review File · Nature Communications]

Supplementary Information for

Ultrafast probes of electron-hole transitions between two atomic layers

Wen et al.

Supplementary Information for

Ultrafast probes of electron-hole transitions between two atomic layers

Xiewen Wen^{2†}, Hailong Chen^{4†}, Tianmin Wu^{6†}, Zhihao Yu^{1†}, Qirong Yang¹, Jingwen

Deng¹, Zhengtang Liu¹, Xin Guo¹, Jianxin Guan¹, Xiang Zhang², Yongji Gong²,

Jiangtan Yuan², Zhuhua Zhang², Chongyue Yi⁵, Xuefeng Guo¹, Pulickel M. Ajayan²,

Wei Zhuang^{3*}, Zhirong Liu^{1*}, Jun Lou^{2*}, Junrong Zheng^{1*}

1 College of Chemistry and Molecular Engineering, Beijing National Laboratory

for Molecular Sciences, Peking University, Beijing 100871, China

2 Department of Materials Science and NanoEngineering, Rice University, 6100

Main Street, Houston, Texas 77005-1892, United States

3 State Key Laboratory of Structural Chemistry, Fujian Institute of Research on the

Structure of Matter, Chinese Academy of Sciences, Fuzhou, Fujian 350002, China

4 Beijing National Laboratory for Condensed Matter Physics, CAS Key Laboratory

of Soft Matter Physics, Institute of Physics, Chinese Academy of Sciences,

Beijing 100190, China

5 Department of Chemistry, Rice University, 6100 Main Street, Houston, Texas

77005-1892, United States

6 Department of Chemical Physics, University of Science and Technology of China,

Hefei, Anhui 230026, China

†These authors contributed equally to the work

*To whom correspondence should be addressed: junrong@pku.edu.cn,

zhengjunrong@gmail.com, jlou@rice.edu, wzhuang@fjirms.ac.cn,

liuzhirong@pku.edu.cn

Supplementary Figures

**Supplementary Figure 1.** *Waiting time dependent IR spectra of monolayer graphene*
 *and heterostructure after excitation with 1.03 eV photons. (A) Graphene; (B)*
 *heterostructure; (C) pump/probe data detected at 1780 cm⁻¹; (D) pump/probe data*
 *detected at 2194 cm⁻¹. Dots are data, and lines are calculations. Calculation*
 *parameters: graphene Fermi level $\mu = -0.19$ eV, phonon fraction $f_{SCOPs} = 0.15$,*
 *photon flux absorbed $F = 0.0069$ J/m², pump/probe response time $t = 170$ fs, and*
 *electron-phonon coupling parameter $\eta = 6.0 \frac{eV}{\text{\AA}}$. These parameters are used for all*
 *calculations for graphene monolayer excited with 1.03 eV.*

**Supplementary Figure 2.** *Spectrum of heterostructure at $t = 0$ ps after excitation*

with 1.03 eV photons. Dots are data, and the line is calculation. Calculation
 parameters: graphene Fermi level $\mu = -0.17$ eV, phonon fraction $f_{SCOPs} = 0.15$,
 photon flux absorbed $F = 0.0046$ J/m², response time $t = 170$ fs, and
 electron-phonon coupling parameter $\eta = 6.0 \frac{eV}{\text{\AA}}$.

**Supplementary Figure 3.** (A) Calculated electronic population relaxation in
 heterostructure after 1.03 eV excitation. (B) Calculated temperature of graphene in
 heterostructure. (C) Calculated transient IR signal (2185 cm⁻¹) of graphene in
 heterostructure after 1.03 eV excitation. Calculation parameters: graphene Fermi
 level $\mu = -0.17$ eV, phonon fraction $f_{SCOPs} = 0.15$, photon flux absorbed
 $F = 0.0046$ J/m², pump response time $t = 170$ fs, and electron-phonon coupling
 parameter $\eta = 6.0 \frac{eV}{\text{\AA}}$. Subtracting (C) from the experimental signal of
 heterostructure (Supplementary Figure 3E) yields the excitonic signal (Supplementary
 Figure 3F). Here the temperature increase from energy released from exciton
 formation is ignored. The signal reduction because of exciton formation is not
 considered either. The effects of these two factors on the graphene signal are opposite.
 Because of this, we estimate the uncertainty of the exciton formation time introduced
 by the approximations to be <20% (on average around 25% free carriers form

excitons). The exciton formation rate is slightly underestimating by this method.

**Supplementary Figure 4.** Excitonic signal of heterostructure after 1.03 eV excitation.

The data (dots) are obtained by directly subtracting the graphene signal from the

heterostructure signal. Calculations (line) show that the excitonic formation time is

325 fs, which is essentially the same as 364 fs of Supplementary Figure 3F. Both time

constants are not normalized to the absorption cross section. The line is a

biexponential with $t_1=120$ fs (electronic decay time constant from graphene

calculations) with factor -0.37, and $t_2=55$ ps with factor 0.37. In Supplementary

Figure 3F, the calculation for excitonic signal uses parameters: $t_1=120$ fs (electronic

decay time constant from graphene calculations) with factor -0.33, and $t_2=65$ ps with

factor 0.33.

**Supplementary Figure 5.** *Calculated graphene 400nm-excitation/2185cm⁻¹-detection*
 *electronic dynamics (A) and signals (B) with two different excitation fluxes.*

*Calculation parameters: graphene Fermi level $\mu = -0.17$ eV , phonon*
 *fraction $f_{SCOPs} = 0.15$, photon flux absorbed*

*$F = 0.0087 \frac{J}{m^2}$ (100%flux) and $0.0261 \frac{J}{m^2}$, pump response time $t = 170$ fs, and*
 *electron-phonon coupling parameter $\eta = 6.0 \frac{eV}{\text{\AA}}$. The electronic decay time is 130fs*

*with 100% and 170fs with 300% flux.*

**Supplementary Figure 6.** *Comparisons of calculations with different electronic*
 *dephasing linewidths and times of graphene. Calculation parameters: graphene*

*Fermi level $\mu = -0.17$ eV, phonon fraction $f_{SCOPs} = 0.20$, $f_{SCOPs} = 0.15$, $f_{SCOPs} =$*
 *0.1 for calculations with $\Gamma = 80$ cm⁻¹ and $\tau_e = 25$ fs, $\Gamma = 500$ cm⁻¹ and $\tau_e = 10$ fs, $\Gamma =$*

1000 cm^{-1} and $\tau_e = 5 \text{ fs}$, respectively; photon flux absorbed
$F = 0.0087 \frac{\text{J}}{\text{m}^2}$ (100% flux), pump response time $t = 170 \text{ fs}$, and electron-phonon
coupling parameter $\eta = 6.0 \frac{\text{eV}}{\text{\AA}}$. The parameters only slightly affect the linshape and
dynamics.

**Supplementary Figure 7.** Temperature dependent n_h/n_e

**Supplementary Figure 8.** Band structure. Electronic band structures of boron-doped

graphene with different primitive cell.

**Supplementary Figure 9.** (a) Fermi energy level of boron-doped graphene with

different primitive cell. (b) The effective mass of B-doped system for electron and hole

transport with different primitive cell. As the Fermi energy level is -0.17 eV, the

primitive cell size is 26.1×26.1 , and the corresponding hole and electron effective

mass is 0.0121 and 0.0116, respectively.

**Supplementary Figure 10.** The calculated Electronic band structures of monolayer

*MoSe₂*, underestimates the band gap by around 0.5 eV.

**Supplementary Figure 11.** Also Supplementary Figure 3D in main text. A few

eigenvalues are shown on the Model Coulomb potential as a function of in-plane

radius for an electron-hole pair across the graphene/*MoSe₂* heterojunction van der

Waals interface. (a) Hole is in graphene and electron is in the *MoSe₂* monolayer. (b)

Hole is in MoSe₂ and electron is in graphene. Wave functions of several states are
 also shown. (red/blue: positive/negative.)

**Supplementary Figure 12.** The exciton transition energy ($1s-2p$) of graphene/MoSe₂
 correlated with effective mass of the excitonic quasi particle.

**Supplementary Figure 13.** (a) Fermi energy level of boron-doped graphene with
 different primitive cell. (b) The effective mass of B-doped system for electron and hole

transport with different primitive cell. When the Fermi energy level is -0.19 eV, the
 primitive cell size is 19.4×19.4 , and the corresponding hole and electron effective
 mass is 0.0132 and 0.0122 , respectively.

**Supplementary Figure 14.** A few eigenvalues are shown on the Model Coulomb
 potential as a function of in-plane radius for an electron-hole pair across the
 graphene/MoSe₂ heterojunction van der Waals interface. **(a)** Hole is in graphene and
 electron is in MoSe₂ monolayer. **(b)** Hole is in MoSe₂ slab and electron is in graphene.
 Several wave functions are also shown. (red/blue: positive/negative.) Fermi level
 164 -0.19 eV.

graphene_hole/MoSe₂_electron: $E_{1s-2p} = 0.287$ eV, $\langle \rho_{CT_{1s}} \rangle = 43.1$ Å

graphene_electron/MoSe₂_hole: $E_{1s-2p} = 0.276$ eV, $\langle \rho_{CT_{1s}} \rangle = 45.1$ Å

**Supplementary Figure 15.** *Band structure. Electronic band structures of*
 *aluminum-doped graphene with different primitive cell.*

**Supplementary Figure 16.** *Fermi energy level of aluminum-doped graphene with*
 *different primitive cell.*

**Supplementary Figure 17.** The effective mass of B-doped system for electron and
hole transport with different primitive cell. As the Fermi energy level is -0.17 eV, the
primitive cell size is 13×13 , and the corresponding hole and electron effective mass is
0.0122 and 0.0135 , respectively.

**Supplementary Figure 18.** A few eigenvalues are shown on the Model Coulomb

potential as a function of in-plane radius for an electron-hole pair across the
 graphene/MoSe₂ heterojunction van der Waals interface. (a) Hole is in graphene and
 electron is in the MoSe₂ monolayer. (b) Hole is in MoSe₂ and electron is in graphene.
 Wave functions of several states are also shown. (red/blue: positive/negative.)

 **Supplementary Figure 19.** A few eigenvalues are shown on the Model Coulomb
 potential as a function of in-plane radius for an electron-hole pair across the
 graphene/MoSe₂ heterojunction van der Waals interface. (a) Hole is in graphene and
 electron is in MoSe₂ monolayer. (b) Hole is in MoSe₂ slab and electron is in graphene.
 Several wave functions are also shown. (red/blue: positive/negative.)

graphene_hole/MoSe₂_electron: $E_{1s-2p} = 0.273 \text{ eV}$, $\langle \rho_{CT_{1s}} \rangle = 46.4 \text{ \AA}$

graphene_electron/MoSe₂_hole: $E_{1s-2p} = 0.274 \text{ eV}$, $\langle \rho_{CT_{1s}} \rangle = 45.4 \text{ \AA}$

**Supplementary Figure 20.** Raman spectroscopy of graphene, MoSe₂ and
 graphene/MoSe₂ vdW heterostructure. The frequencies of vibrational modes A_{1g} (242
 201 cm⁻¹) and E_{2g}¹ (288 cm⁻¹) of the standalone MoSe₂ sample and the heterostructures
 match previously reported data of monolayer MoSe₂²⁶. The characteristic 2D (2690
 203 cm⁻¹) and G (1580 cm⁻¹) peaks of single-layered graphene are present in both the
 204 graphene/MoSe₂ heterostructures and the standalone graphene sample.

**Supplementary Figure 21.** *Schematic illustration of the finite effective mass when k*
 *deviates away from the Dirac point of graphene. Some k -lines on the Dirac cones*
 *along a specified direction were shown as solid lines, which clearly have finite*
 *curvature (effective mass).*

**Supplementary Figure 22.** *The effective mass m^* (in units of m_0 , the mass of the free*
 *electron) at the level of chemical potential μ along the normal direction of Dirac cone*

in doped graphene. Supplementary Equation (24) is used in the plot.

**Supplementary Figure 23.** The exciton binding energy as a function of the effective
mass m^* (in units of m_0 , the mass of the free electron). The binding energy was
measured with respect to that of MoSe_2 ($m^* = 0.6$). Supplementary Equation (26)
and (28) were used in plotting the curves.

**Supplementary Figure 24.** (A) Detection frequency dependent (0.28-0.59 eV) photo
responses of monolayer MoSe_2 at 10 ps (averaged between 8-12 ps to minimize noise)
after excited with 400nm light. A resonance $1s$ - $2p$ transition peak appears at 0.55 eV.
The results indicate that the excitonic binding energy in monolayer MoSe_2 is about 0.6
228 eV. (B) Time dependent dynamics detected at different energy values. The transition
between different dynamics starts at 0.51eV. The DFG midIR output from Topas OPA

combined with HgCdTe detector was used for detection frequencies below 0.47 eV. The
Idler output of a Palitra OPA combined with an InSb detector was used for detection
frequencies above 0.47 eV.

**Supplementary Figure 25.** The band structure of a graphene/MoSe₂ heterostructure.

The symmetry of graphene is broken, and its band structure is opened with a gap of ~

4.4 meV (the insert). This generates a carrier mass around 0.0004 m_0 . This

phenomenon was also previously observed in calculations on a graphene/MoSe₂

heterostructure (JPCC, 115, 20237, 2011). However, the gap is significantly smaller

than the thermal energy (26 meV) at room temperature, the effects it induced at room

temperature under which our experiments were conducted are expected to be very

small.

**Supplementary Figure 26.** *MoSe₂/Graphene device used in the measurements, scale*
 *bar: 10 μm. MoSe₂/graphene vdW heterostructure device for photocurrent*
 *measurements. A monolayer of graphene (the lighter image in the center) is*
 *transferred on the top of A monolayer of MoSe₂ which is on a silicon wafer. Cr/Au*
 *(8nm/120nm) electrodes were deposited using E-beam lithography and lift-off*
 *technique. Both electrodes are on the top of graphene. One is on the top of graphene*
 *underneath which is Si (with a thin layer of SiO₂). The other electrode is on the top of*
 *graphene underneath which is MoSe₂.*

**Supplementary Figure 27.** *Photocurrent ($I_{ph}=I_{light}-I_{dark}$) of MoSe₂/Graphene vdW*
 *heterostructure under 980nm excitation. The photons only excite graphene but not*
 *MoSe₂. Top panel: measurement configurations (A) in which the external current*
 *flows from graphene on the top of MoSe₂ to graphene on the top of Si (with a thin*
 *layer of SiO₂), and (B) the external current direction is flipped. The background*
 *current is subtracted. Both results unambiguously demonstrate that the*
 *photoexcitation generates an overall electron flow from MoSe₂ to graphene inside the*
 *heterostructure. After photoexcitation, electrons move from MoSe₂ to graphene inside*
 *the heterostructure. Therefore, more electrons exist in the area of graphene directly on*
 *the top of MoSe₂ than in the area of graphene on the top of Si (with a thin layer of*

SiO_2). This results in external current flowing from graphene on the top of Si to
 graphene on the top of MoSe_2 . The photocurrent measurements were conducted with
 a semiconductor parameter analyzer (Agilent 4155C) and a probe station. The
 source/drain bias $V_{sd} = 0$ and the gate voltage $V_g = 0$ in all tests. The two
 measurements were conducted under slightly different laser focus sizes.

 **Supplementary Figure 28.** Photocurrent ($I_{ph} = I_{light} - I_{dark}$) of $\text{MoSe}_2/\text{Graphene}$ vdW
 heterostructure under 405nm excitation. The photons excite both graphene and MoSe_2 .
 Top panel: measurement configurations (A) in which the external current flows from
 graphene on the top of MoSe_2 to graphene on the top of Si (with a thin layer of SiO_2),
 and (B) the external current direction is flipped. The background current is subtracted.
 Both results unambiguously demonstrate that the photoexcitation generates an overall
 electron flow from MoSe_2 to graphene inside the heterostructure. After
 photoexcitation, electrons move from MoSe_2 to graphene inside the heterostructure.
 Therefore, more electrons exist in the area of graphene directly on the top of MoSe_2
 than in the area of graphene on the top of Si (with a thin layer of SiO_2). This results in
 external current flowing from graphene on the top of Si to graphene on the top of
 MoSe_2 . The photocurrent measurements were conducted with a semiconductor
 parameter analyzer (Agilent 4155C) and a probe station. The source/drain bias V_{sd}
 $= 0$ and the gate voltage $V_g = 0$ in all tests. Similar photocurrent phenomena was also
 observed in $\text{MoS}_2/\text{Graphene}$ vdW heterostructures (ref: Scientific Reports 4, 3826
 (2014)).

Supplementary Notes

**Supplementary Note 1: Optical conductivity of graphene**

The tight-binding Hamiltonian for the cone-like band structure of graphene is
written as

$$293 \quad H = \begin{bmatrix} 0 & \hbar v_F (k_x + ik_y) \\ \hbar v_F (k_x - ik_y) & 0 \end{bmatrix}, \quad (1)$$

where v_F is the Fermi velocity, and k_x and k_y are 2D components of the electronic
wave vector \mathbf{k} . With the Hamiltonian, the optical conductivity $[\sigma(\omega)]$ is written as the
sum of the interband conductivity $[\sigma_{\text{inter}}(\omega)]$ and the intraband conductivity $[\sigma_{\text{intra}}(\omega)]$
given below^{1, 2, 3, 4}:

$$\begin{aligned} \sigma(\omega) &= \sigma_{\text{inter}}(\omega) + \sigma_{\text{intra}}(\omega) \\ 298 \quad \sigma_{\text{inter}}(\omega) &= i \frac{e^2 \hbar \omega}{\pi \hbar} \int_0^{+\infty} d\varepsilon \frac{1}{(2\varepsilon)^2 - (\hbar\omega + i\Gamma)^2} [f_{\text{FD}}(\varepsilon - \mu) - f_{\text{FD}}(-\varepsilon - \mu)], \quad (2) \\ \sigma_{\text{intra}}(\omega) &= i \frac{e^2 / \pi \hbar}{\hbar\omega + i\hbar/\tau_e} \int_0^{+\infty} d\varepsilon [f_{\text{FD}}(\varepsilon - \mu) + 1 - f_{\text{FD}}(-\varepsilon - \mu)] \end{aligned}$$

where f_{FD} is the Fermi-Dirac distribution function, μ is the chemical potential (Fermi
energy), and e is the elementary charge. Γ is the broadening of the interband
transitions, and τ_e is the relaxation time due to intraband carrier scattering. In
literature^{5, 6}, Γ lies between 0.01 and 0.06 eV while τ_e is in the range of 5 ~ 40 fs.
Here, considering that electronic motions typically occur within a few fs rather tens of
fs, we use the fast values $\Gamma = 0.062$ eV (500 cm^{-1}) and $\tau_e = 10$ fs throughout the study.
Nonetheless, effects of these two parameters are also tested and listed in
Supplementary Figure 6. It turns out that the results are not very sensitive to the
parameter selections.

Applying the Fresnel equations, the change of optical transmission due to the

existence of graphene is given as

$$310 \quad \frac{\Delta T_s}{T_0} \approx -\frac{2}{\cos \theta + n_{\text{sub}} \cos \theta''} \sqrt{\frac{\mu_0}{\epsilon_0}} \text{Re}[\sigma(\omega)] \quad (3)$$

for s-polarized light. θ is the incident angle, and θ'' is the incident angle in the
 substrate. n_{sub} is the refractive index of the substrate. ϵ_0 and μ_0 are vacuum
 permittivity and permeability, respectively. For p-polarized light, the change of optical
 transmission is

$$315 \quad \frac{\Delta T_p}{T_0} \approx -\frac{2 \cos \theta \cos \theta''}{n_{\text{sub}} \cos \theta + \cos \theta''} \sqrt{\frac{\mu_0}{\epsilon_0}} \text{Re}[\sigma(\omega)]. \quad (4)$$

In our experiments, $\theta = 0^\circ$, so we have

$$317 \quad \frac{\Delta T}{T_0} \approx -\frac{2}{1 + n_{\text{sub}}} \sqrt{\frac{\mu_0}{\epsilon_0}} \text{Re}[\sigma(\omega)], \quad (5)$$

where $n_{\text{sub}} = 1.39$ for the used substrate (CaF₂).

**Supplementary Note 2: Heat transfer between electrons and phonons in**
 **graphene**

Strongly coupled optical phonons (COPs) are in-plane optical phonons for which
 intra- and inter-valley carrier scattering can simultaneously conserve energy and
 momentum⁷. The considered SCOPs include phonons near the Γ -point with energy \sim
 200 meV (for intra-valley carrier scattering) and those near the K -point with energy
 \sim 150 meV (for inter-valley carrier scattering). The interaction between phonons and
 electrons/holes in graphene is described by a deformation potential theory,^{8,9} and the
 transition matrix element is determined to be

$$M_{\mathbf{k}',\mathbf{k}}^{(\text{TO}\&\text{LO})} \approx 3\eta \sqrt{\frac{\hbar}{4M_C \omega_{\text{phonon}}}}, \quad (6)$$

where η is the electron-phonon coupling parameter, and M_C is the mass of a carbon atom. The energy dispersion of the transverse optical (TO) and longitudinal optical (LO) phonon modes is ignored, and Supplementary Equation (6) accounts for the total contribution from TO and LO at Γ - or K -points. Both emission and absorption of the phonons are considered under the second quantization. The probability of scattering from \mathbf{k} to \mathbf{k}' by the SCOPs is

$$\begin{aligned} W_{\mathbf{k}',\mathbf{k}}^{(\text{TO}\&\text{LO})} &= \frac{2\pi}{\hbar} \left| M_{\mathbf{k}',\mathbf{k}}^{(\text{TO}\&\text{LO})} \right|^2 \left[N_{\mathbf{q}} \delta(\varepsilon_{\mathbf{k}} - \varepsilon_{\mathbf{k}'} + \hbar\omega_{\text{phonon}}) + (N_{\mathbf{q}} + 1) \delta(\varepsilon_{\mathbf{k}} - \varepsilon_{\mathbf{k}'} - \hbar\omega_{\text{phonon}}) \right] \\ &= \frac{9\pi\eta^2}{2M_C \omega_{\text{phonon}}} \left[N_{\mathbf{q}} \delta(\varepsilon_{\mathbf{k}} - \varepsilon_{\mathbf{k}'} + \hbar\omega_{\text{phonon}}) + (N_{\mathbf{q}} + 1) \delta(\varepsilon_{\mathbf{k}} - \varepsilon_{\mathbf{k}'} - \hbar\omega_{\text{phonon}}) \right], \end{aligned} \quad (7)$$

where $N_{\mathbf{q}}$ is the phonon occupation number described by the Bose-Einstein distribution. The first and second terms in the square brackets correspond to the phonon emission and absorption processes, respectively. Integrating \mathbf{k}' gives the scattering rate of \mathbf{k} as

$$\begin{aligned} R_{\mathbf{k} \rightarrow * } &= \int W_{\mathbf{k}',\mathbf{k}}^{(\text{TO}\&\text{LO})} \frac{A}{(2\pi)^2} d^2\mathbf{k}' \\ &= \frac{9\eta^2}{2\hbar^2 \rho v_F^2 \omega_{\text{phonon}}} \left[(\varepsilon_{\mathbf{k}} + \hbar\omega_{\text{phonon}}) N_{\mathbf{q}} + (\varepsilon_{\mathbf{k}} - \hbar\omega_{\text{phonon}}) (N_{\mathbf{q}} + 1) \right], \end{aligned} \quad (8)$$

where ρ is the 2D mass density of graphene. Therefore, the rate of phonon emission per unit area is

$$\begin{aligned} \frac{dN_{\text{emission}}}{dt} &= \int \frac{9\eta^2}{2\hbar^2 \rho v_F^2 \omega_{\text{phonon}}} (\varepsilon_{\mathbf{k}} - \hbar\omega_{\text{phonon}}) (N_{\mathbf{q}} + 1) f_{\text{FD}}(\varepsilon_{\mathbf{k}}) (1 - f_{\text{FD}}(\varepsilon_{\mathbf{k}} - \hbar\omega_{\text{phonon}})) \frac{1}{(2\pi)^2} d^2\mathbf{k} \\ &= \frac{9\eta^2}{4\pi(\hbar v_F)^4 \rho \omega_{\text{phonon}}} (N_{\mathbf{q}} + 1) \int \varepsilon_{\mathbf{k}} (\varepsilon_{\mathbf{k}} - \hbar\omega_{\text{phonon}}) f_{\text{FD}}(\varepsilon_{\mathbf{k}}) (1 - f_{\text{FD}}(\varepsilon_{\mathbf{k}} - \hbar\omega_{\text{phonon}})) d\varepsilon_{\mathbf{k}} \end{aligned}$$

(9)

and the rate of phonon absorption is

$$348 \quad \frac{dN_{\text{adsorption}}}{dt} = \frac{9\eta^2}{4\pi(\hbar v_F)^4 \rho \omega_{\text{phonon}}} N_q \int \epsilon_{\mathbf{k}} (\epsilon_{\mathbf{k}} + \hbar \omega_{\text{phonon}}) f_{\text{FD}}(\epsilon_{\mathbf{k}}) (1 - f_{\text{FD}}(\epsilon_{\mathbf{k}} + \hbar \omega_{\text{phonon}})) d\epsilon_{\mathbf{k}},$$

(10)

which are used to simulate the heat transfer between electrons/holes and SCOPs.

The heat capacity of the SCOPs is described by the Einstein model, and the
 fraction of Brillouin zone filled by the SCOPs, f_{SCOPs} , is assumed to be temperature
 independent for simplicity⁴. The energy stored in the SCOP subsystem relaxes at a
 rate of $1/\tau_{\text{ph}}$ to lower energy phonons^{4,10}. The value of f_{SCOPs} and τ_{ph} are determined
 by fitting the experimental probe signal profiles.

**Supplementary Note 3: Fermi level of graphene in heterostructure**

Graphene used in our experiments are p-doped with Fermi level -0.19 eV
 determined by fitting the transient spectra and dynamics in Supplementary Figure 1
 and Supplementary Figure 3E. MoSe₂ is n-doped. When the two monolayers were
 placed together to form a heterostructure, some electrons transfer from MoSe₂ to
 graphene¹¹, raising the graphene Fermi level to -0.17 eV. The Fermi level of graphene
 in heterostructure is obtained by fitting the transient IR spectrum of heterostructure at
 time 0 with 1.03 eV excitation (Supplementary Figure 2). The carrier density of
 graphene is

$$366 \quad n = \int \frac{1}{(2\pi)^2} f(\epsilon - \mu) d^2\mathbf{k} = \int \frac{k}{2\pi} f(\epsilon - \mu) dk = \int_0^{+\infty} \frac{\epsilon}{2\pi(\hbar v_F)^2} f(\epsilon - \mu) d\epsilon, \quad (11)$$

When $|\mu| \gg k_B T$, Supplementary Equation 11 can be simplified into (degenerate spin
and KK'):

$$369 \quad n \approx \frac{\mu^2}{\pi(\hbar v_F)^2}. \quad (12)$$

According to Supplementary Equation 11&12, the Fermi level increase indicates
that about $5 \times 10^{11} \frac{e}{cm^2}$ have transferred from MoSe₂ to graphene after they form a
heterostructure.

**Supplementary Note 4: Calculated IR response of graphene with Fermi level**
**-0.17 eV**

In the heterostructure excited with 1.03 eV photons, the initial response is only
from graphene. Because of fast interlayer charge transfers, some excited carriers
move to MoSe₂ very rapidly (< 50 fs) and effectively reduce the electronic
temperature in the graphene layer. In our calculations, we consider this problem
equivalent to the reduction of excitation flux because the charge transfer and
electronic thermalization are extremely fast (< 50 fs), much faster than the exciton
formation and electron/phonon couplings. Under ideal fast equilibrium, at most 50%
of the excitation energy is transferred to MoSe₂ by the transferred carriers. However,
in our experiments, this is obviously not likely because very few electrons can reach
CBM of MoSe₂. Therefore, we set the upper limit of flux reduction as 1/3. Calculation
results for the exciton formation time with no flux reduction to 1/3 reduction are all
within experimental uncertainty.

**Supplementary Note 5: Excitonic signal from direct experimental data**
**subtraction**

The excitonic signal presented in Supplementary Figure 3F is obtained by
considering pump flux and Fermi level changes. Since these two parameters don't
change significantly, we also directly subtract the signal of monolayer graphene from
that of heterostructure (both presented in Supplementary Figure 3E) to obtain the
excitonic signal. Kinetic analyses on the result by this method are plotted in
Supplementary Figure 4. The results from this method and that from the more
accurate method in Supplementary Figure 3F turn out to be essentially the same.

**Supplementary Note 6: Larger flux results in slower graphene signal decay**

MoSe₂ absorbs 14% and graphene absorbs 4.14% of 3.1 eV photons. Under ideal
fast equilibrium, ~9% of excitation photons would be in either layer of the
heterostructure. However, because the graphene Fermi level is between MoSe₂ VBM
and CBM, more carriers are expected to flow into graphene. From photoluminescence,
we can see that around 15% of free carriers that are originally excited in MoSe₂
remain in MoSe₂. Estimated from the results with 1.03 eV photons, about 33% free
carriers form interlayer excitons. These two portions together account for 6% of the
18.14% total absorption. Therefore, the excitation flux the graphene layer gains from
interlayer charge transfers must not exceed 12.5% (300% of 4.14%) besides direct
absorption. The charge transfers results in higher electronic temperature and slower
dynamics in graphene.

**Supplementary Note 7: MoSe₂ bandgap**

According to STM measurements¹², the bandgap of MoSe₂ monolayer samples
on different substrates varies. On graphite, it is 1.94 eV, whereas on bilayer graphene
it is 2.18 eV. The PL peak central frequency also varies from 1.67 eV to 1.63 eV at
77K. The determined binding energy is 0.55 eV, slightly higher than the theoretical
value 0.47 eV¹³. The PL central frequency of our sample at room temperature is 1.56
418 eV (on graphene). 1.56 + 0.47 = 2.03 eV. Considering all results from literature and
419 our own experiments, we adopt around 2.0 eV as the bandgap of MoSe₂ in the
420 heterostructures measured in our experiments.

**Supplementary Note 8: Ratio of electron/hole transferred from graphene to**
**MoSe₂**

The population of electrons in graphene that have higher energy than the CBM
value of MoSe₂ and thus are capable of interlayer transferring is calculated as

$$n_e(T; E_{\text{CBM}}) = \int_{E_{\text{CBM}}}^{+\infty} \frac{\mathcal{E}}{2\pi(\hbar v_F)^2} f(\mathcal{E} - \mu) d\mathcal{E}. \quad (13)$$

Similarly, the population of holes being capable of interlayer transferring is

$$n_h(T; E_{\text{VBM}}) = \int_{-\infty}^{E_{\text{VBM}}} \frac{\mathcal{E}}{2\pi(\hbar v_F)^2} f(-\mathcal{E} + \mu) d\mathcal{E}. \quad (14)$$

By subtracting the populations before pumping ($T_0=300\text{K}$), the ratio of n_h/n_e
determined is plotted in Supplementary Figure 7. With 100% flux of 1200nm
excitation, the electronic temperature reaches 1583K. At this temperature, n_h/n_e is 4.9.

With 2/3 flux, the electronic temperature reaches 1399 K and n_h/n_e is 5.8. Because
charge transfers result in a lower temperature, the exact n_h/n_e should lie between these
two values. Interlayer charge transfers and electronic thermalization in graphene are
extremely fast (<50 fs)^{4, 14, 15}. Thus, they can be treated as semi-instantaneous,
compared to the electron/hole recombination (100~200 fs) in graphene and the
formation of interlayer excitons (~500 fs). Therefore, the transfer of photo-excited
charge carriers between two layers forms a quasi-equilibrium in which the
electron/hole population ratio in MoSe₂ is approximately equal to that in graphene.
Once the charge carriers form interlayer excitons, the interlayer quasi-equilibrium is
perturbed and more carriers are transferred to MoSe₂ as the charge carriers in
graphene are continuously adjusting their population distributions. This process
makes sure that even though the hole population in graphene below the VBM of
MoSe₂ is only a small portion of the total number photo-generated, the total number
of transferred holes is still substantial. The process results in that the majority of
interlayer excitons have their holes in the MoSe₂ layer and electrons in graphene. The
estimated exciton density is about $0.45 \times 10^{12} \text{ cm}^{-2}$ and $0.58 \times 10^{12} \text{ cm}^{-2}$ under the
excitation photon energy 1.03 eV and 3.1 eV, respectively. Another possible
mechanism can also lead to that electrons stay in graphene and holes prefer MoSe₂.
According to literature¹¹, charged impurities in MoSe₂ and graphene can produce an
effective electric field that is from graphene to MoSe₂. Such an electric field can
provide driving force for holes to move to MoSe₂ and for electrons to go to graphene.

**Supplementary Note 9: Kinetic analysis**

To analyze the electron/hole gas transformation dynamics, we create a kinetic
 model. In the model, interlayer charge transfers and electronic thermalization in
 graphene are assumed to be semi-instantaneous. Free carriers (y) in graphene decay
 with a total rate constant k_3 . The interlayer exciton (x) formation rate constant is k_1
 and the exciton decay rate constant is k_2 . The model can be expressed as

The solution of the kinetic model is

$$\begin{aligned} \frac{dx}{dt} &= k_1[y] - k_2[x] \\ \frac{dy}{dt} &= -k_3[y] \end{aligned} \quad , \quad (16)$$

$$x(t) = \frac{k_1}{k_3 - k_2} (1 - e^{-(k_3 - k_2)t}) e^{-k_2 t}$$

where k_3 is obtained from the electronic decay of graphene. k_1 and k_2 are
 obtained from fitting the experimental excitonic signal.

In Supplementary Figure 3F, $1/k_1 = 364 \text{ fs}$, $1/k_2 = 65 \text{ ps}$, $1/k_3 = 120 \text{ fs}$.

In Supplementary Figure 4F, $1/k_1 = 379 \text{ fs}$, $1/k_2 = 55 \text{ ps}$, $1/k_3 = 170 \text{ fs}$.

The binding energy of MoSe₂ intralayer exciton is >0.4 eV, which exceeds the
 photon energy of our IR probe. In addition, the optical response of free carriers in
 MoSe₂ is only 1/6 of the free carriers in graphene. Therefore, in the kinetic model the
 contribution from free carriers in MoSe₂ is ignored. The response ratio 1/6 is
 calculated based on experimental results: with the same 3.1 eV excitation, 14%
 photons are absorbed by MoSe₂, producing a signal size of -0.006, whereas 4.14%

photons are absorbed by graphene producing a signal size of -0.011.
 $0.011/0.006*14\%/4.14\% \sim 6$.

**Supplementary Note 10: Exciton and free carrier optical responses**

The free carrier optical conductivity from the Drude model¹⁶ is

$$478 \quad \sigma_{free} = \frac{ne^2\tau}{\mu(\omega^2\tau^2 + 1)} = \frac{n\hbar e^2\Gamma_D}{\mu[(\hbar\omega)^2 + \Gamma_D^2]}, \quad (17)$$

where n is the carrier population, μ is the reduced mass and Γ_D is the scattering
 (dephasing) line width. The 1s-2p transition optical conductivity¹⁷ is

$$481 \quad \sigma_{1s-2p} = \frac{n_X \hbar e^2 \Gamma_{1s-2p} \cdot (\hbar\omega)^2 f_{1s-2p}}{\mu \left[((\hbar\omega)^2 - E_{res}^2)^2 + (\hbar\omega \Gamma_{1s-2p})^2 \right]}, \quad (18)$$

where E_{res} is the resonant energy. The optical conductivity ratio between the 1s-2p
 transition and free carrier is therefore:

$$484 \quad \frac{\sigma_{1s-2p}}{\sigma_{e-h}} \approx \frac{2n_X f_{1s-2p}}{\Gamma_{1s-2p}} \bigg/ \frac{n_{e-h} \Gamma_D}{(\hbar\omega)^2} \approx \frac{2f_{1s-2p} (\hbar\omega)^2}{\Gamma_D \Gamma_{1s-2p}}, \quad (19)$$

where f_{1s-2p} is the oscillator strength that can be calculated in the following¹⁸:

$$486 \quad \begin{aligned} f_{1s-n} &= \frac{2\mu a^2}{\hbar^2} (E_n - E_0) \frac{\left(\frac{n+1}{2}\right)^5 n^{2n-3}}{(n+1)^{2n+5}} \\ &= \frac{2\mu a^2}{\hbar^2} \cdot \frac{e^2 \lambda}{8\pi \epsilon_0 \epsilon_r a} \left[\frac{1}{(1/2)^2} - \frac{1}{(n+1/2)^2} \right] \frac{\left(\frac{n+1}{2}\right)^5 n^{2n-3}}{(n+1)^{2n+5}} \\ &= \frac{\mu e^2 \lambda a}{4\pi \epsilon_0 \epsilon_r \hbar^2} \left[\frac{1}{(1/2)^2} - \frac{1}{(n+1/2)^2} \right] \frac{\left(\frac{n+1}{2}\right)^5 n^{2n-3}}{(n+1)^{2n+5}} \\ &= \frac{\mu e^2 \lambda}{4\pi \epsilon_0 \epsilon_r \hbar^2} \cdot \frac{4\pi \hbar^2 \epsilon_0 \epsilon_r}{e^2 \mu \lambda} \left[\frac{1}{(1/2)^2} - \frac{1}{(n+1/2)^2} \right] \frac{\left(\frac{n+1}{2}\right)^5 n^{2n-3}}{(n+1)^{2n+5}} \\ &= \left[\frac{1}{(1/2)^2} - \frac{1}{(n+1/2)^2} \right] \frac{\left(\frac{n+1}{2}\right)^5 n^{2n-3}}{(n+1)^{2n+5}} \end{aligned} \quad (20)$$

Based on Supplementary Equation 18, $f_{1s-2p} = 0.21$, slightly different from the
 value 0.32 derived from a non-hydrogen model¹⁷. Γ_{1s-2p} is experimentally determined

to be 280 cm^{-1} . Using the ratio $\Gamma_D / \Gamma_{1s-2p} = 6$ adopted in literature¹⁸, $\frac{\sigma_{1s-2p}}{\sigma_{e-h}}$ is
calculated to be 3.6. However, $\frac{\sigma_{1s-2p}}{\sigma_{e-h}}$ calculated here is for exciton and free carrier in
the same material. In our experiments, the interlayer exciton is within two very
different materials of which the free carrier response ratio is $1/6 = \text{MoSe}_2/\text{graphene}$. It
is reasonable to expect that σ_{1s-2p} of the $\text{MoSe}_2/\text{graphene}$ interlayer exciton is 1~3.5
494 times (it doesn't go to 6 because the 1s-2p transition is a bound-bound transition and
495 graphene cannot affect it in a way like free carriers) of σ_{1s-2p} of MoSe_2 , giving
$\frac{\sigma_{1s-2p\text{-interlayer}}}{\sigma_{e-h\text{-graphene}}} = 0.6 \sim 2.1$. Normalized with the ratio, the excitonic formation time
constants obtained from fitting the experimental results in Supplementary Figure
3F&4F are 218 fs~764 fs (Supplementary Figure 3F) and 227 fs~796 fs
(Supplementary Figure 4F), respectively.

**Supplementary Note 11: Exciton 1s-2p transition energy calculations**

Methods:

First-principle calculations in this work are conducted using DFT methods
implemented in the Vienna *ab initio* simulation package (VASP)¹⁹. The
projector-augmented-wave (PAW) pseudopotentials and the generalized gradient
approximation of Perdew, Burke, and Ernzerhof (PBE) for exchange-correlation
functional are adopted in our simulations²⁰. In order to study the influence of Fermi
level on the electronic properties for hole-doped graphene system, we conducted the
first-principle investigation on the electronic band structures of graphene sheets

($N \times N \times 1$ primitive unit cell) doped by the Boron atoms (which can effectively
modulate the position of Fermi level but minimally alter other physical properties of
graphene). For Brillouin zone (BZ) integrations, a Monkhorst-Pack k-point mesh
scheme with $7 \times 7 \times 1$ are adopted²¹. Meanwhile, for graphene and monolayer MoSe₂
system, we use a $9 \times 9 \times 1$ k-meshes for Brillouin zone sampling in the
Monkhorst-Pack scheme. In order to simulate the graphene/MoSe₂ 2D
heterostructures, we impose a commensurability condition between them, where a
4×4 supercell of the graphene is used to match a 3×3 supercell of the MoSe₂
monolayer. The lateral lattice parameter for the triangular lattice of graphene/MoSe₂
nanocomposite is set to be $a(\text{graphene/MoSe}_2) = (4 \cdot a(\text{graphene}) + 3 \cdot a(\text{MoSe}_2)) =$
9.912 \AA , in which $a(\text{graphene}) = 2.468 \text{ \AA}$ and $a(\text{MoSe}_2) = 3.317 \text{ \AA}$ are optimized for
isolated graphene and monolayer MoSe₂, respectively. Since there is only a very small
lattice mismatch of $\sim 0.4\%$ between the graphene and the MoSe₂ monolayers, the
approximation used here is reasonable. Boron-doped graphene/MoSe₂ nanocomposite
in which one random carbon atom is replaced with a boron atom is also used to study
the influence of doping on the thickness of graphene/MoSe₂ heterojunction (along z
direction). Since the heterostructure contains one graphene and one MoSe₂ monolayer,
van der Waals (vdW) interactions are taken into account by using the semi-empirical
DFT-D3 method²². For the pristine as well as the boron-doped graphene/MoSe₂
heterojunction, a $5 \times 5 \times 1$ Monkhorst-Pack k-point mesh is used. A cutoff of 500 eV
is used for the plane-wave expansion of the wave function to converge the relevant
quantities. The structure relaxations are carried out until all the atomic forces on each

ion are less than $0.01 \text{ eV}/\text{\AA}$, enforcing a total energy convergence of $1 \times 10^{-7} \text{ eV}$. To
avoid spurious interactions with replicas, a vacuum slab larger than 15 \AA is added in
the z direction for all system.

Results:

As demonstrated in Supplementary Figure 8, the Fermi energy levels are adjusted,
by boron doping, to the valence bands and the whole doping system acts as a p-dopant.
Supplementary Figure 9(a) shows the Fermi energy levels of the doped graphene with
different $N \times N$ primitive cells. With the decrease of dopant concentration, the Fermi
energy level gradually decreases to the experimental value. As the Fermi energy level
of boron-doped graphene reaches the experimental value (-0.17 eV), the primitive cell
size is 26.1×26.1 , and the calculated effective mass of B-doped system for electron
and hole is 0.0116 and 0.0121, respectively. (Supplementary Figure 9(a))

The optimized thickness between the two layers (C atom - Mo atom) of
Boron-doped graphene/MoSe₂ heterojunction is 5.178 \AA , which is very close to the
value of undoped heterojunction (5.183 \AA), thereby low concentration of boron doping
does not alter the thickness between the two layers significantly. Furthermore, the
in-plane dielectric constant of B-doped graphene is almost equal to the value of
pristine graphene when the primitive cell size is larger than 8×8 . We set the dielectric
constant of graphene/MoSe₂ heterojunction as $\bar{\epsilon} = 4.969$, which is the average value
of the calculated in-plane dielectric constant $\epsilon_{xx} = 3.651$ and 6.287 for graphene and
monolayer MoSe₂, respectively.

To obtain the potential experienced by an electron at (ρ, z) due to the presence of

a hole at $(0, z_0)$, we use a field method described by Smythe²³ and extended by
 Sritharan¹³:

$$\begin{aligned}
 \quad & V(\rho) = \\
 \quad & -\frac{e^2}{4\pi\epsilon_0\bar{\epsilon}} \left\{ \frac{1}{\sqrt{(z-z_0)^2+\rho^2}} + \sum_{n=0}^{\infty} \left(\frac{\beta_N}{\beta_P} \right)^n \left[\frac{\beta_N/\beta_P}{\sqrt{(z-z_0+2c+2nc)^2+\rho^2}} + \frac{\beta_N/\beta_P}{\sqrt{(z_0-z+2c+2nc)^2+\rho^2}} + \right. \right. \\
 \quad & \left. \left. \frac{(K_2-K_1)/(K_2+K_1)}{\sqrt{(z+z_0-2a+2nc)^2+\rho^2}} + \frac{(K_2-K_3)/(K_2+K_3)}{\sqrt{(2b-z-z_0+2nc)^2+\rho^2}} \right] \right\}, \quad (21)
 \end{aligned}$$

where $\beta_P = (\bar{\epsilon} + 1)^2$ and $\beta_N = (\bar{\epsilon} - 1)^2$, $K_1=K_3=1$ since the graphene/MoSe₂
 heterojunction is approximated as a single dielectric slab sandwiched between
 vacuum. When the electron and hole are at different layers, they are assumed to be at
 the center of the graphene and MoSe₂ monolayer along the z direction with a fixed
 distance of 5.183 Å which is the thickness of heterojunction. The exciton is then
 simplified as a two dimensional quasi-particle²⁴.

When the hole is in graphene and the electron is in the MoSe₂ layer, the effective
 mass of exciton can be obtained from $\frac{1}{\mu} = \frac{1}{m_h^*} + \frac{1}{m_e^*}$, where the effective mass of
 monolayer MoSe₂ conduction band $m_e^* = 0.502m_0$ (Supplementary Figure 10) and
 the effective mass of B-doped graphene valence band $m_h^* = 0.0121m_0$, m_0 is the
 free electron mass. Based on the finite element method, we then solved the
 Schrödinger equation on a 2×10^3 Å circle plane, with the effective mass of the
 excitonic quasi particle and aforementioned Coulomb potential. The eigenvalues and
 selected eigenfunctions calculated are presented in Supplementary Figure 11a. The
 exciton transition energy between 1s and 2p is $E_{1s-2p} = 0.274$ eV, and the means
 radius of wave functions for 1s CT is $\langle \rho_{CT_{1s}} \rangle = 45.3$ Å. The 1s-1p transition energy is
 also larger than 0.20 eV. Both 1s-1p and 1s-2p transition energy values are close to the

experimental value 0.27 eV.

When electron is in graphene and hole is in MoSe₂, the hole and electron
effective masses should be the effective masses of the monolayer MoSe₂ valence
band (0.583 m_0) and B-doped graphene conduction band (0.0116 m_0), respectively.
The eigenvalues and selected eigenfunctions are presented in Supplementary Figure
11b. The exciton transition energy between 1s and 2p is $E_{1s-2p} = 0.267$ eV, and the
mean radius of wave functions for 1s CT is $\langle \rho_{CT_{1s}} \rangle = 46.6$ Å. There is a 0.007 eV
difference between the binding energies in two conditions. In our experiments, the
difference is too small to distinguish.

Note that the energies are calculated for the Fermi level of -0.17 eV, for Fermi
level of -0.19 eV, the calculated values are provided at the end of this summary.

In principle, the exciton 1s-2p transition energy can be estimated by performing
the quasiparticle GW and Bethe-Salpeter equation (BSE) calculations. To model the
experimental condition herein for the boron doped graphene/MoSe₂ heterojunction,
however, the system constructed should contain more than 1200 atoms (the supercell
of graphene should be larger than 26×26). On the other hand, to achieve well-converged
quasiparticle energies, especially when there is a transition metal element (Mo) in the
system, a dense enough ($21 \times 21 \times 1$) k-grid, a large energy cutoff (~ 40 Ry) and a large
number (> 10000) of unoccupied states is needed^{12, 25}. The calculation cost is
therefore far beyond the computing resource we can afford.

**Supplementary Note 12: Al-doped Graphene**

To explore the possible effect of dopant rather than Fermi level on the binding
energy, we also compute the interlayer exciton binding energy with Al-doped
graphene.

As demonstrated in *Supplementary Figure 15*, the Fermi energy levels are
adjusted, by aluminum doping, to the valence bands and the whole doping system also
acts as a p-dopant. *Supplementary Figure 15* shows the Fermi energy levels of the
doped graphene with different $N \times N$ primitive cells. With the decrease of dopant
concentration, the Fermi energy level gradually decreases to the experimental value.
As the Fermi energy level of aluminum-doped graphene reaches the experimental
value (-0.17eV), the primitive cell size is 13×13 (*Supplementary Figure 16*), and the
calculated effective mass of Al-doped system for electron and hole is 0.0122 and
0.0135, respectively. (*Supplementary Figure 17*)

When the hole is in graphene and the electron is in the MoSe_2 layer, the effective
mass of exciton can be obtained from $1/(\mu) = 1/(m_h^*) + 1/(m_e^*)$, where the effective
mass of monolayer MoSe_2 conduction band $m_e^* = 0.502m_0$ (*Supplementary Figure 18*)
and the effective mass of Al-doped graphene valence band $m_h^* = 0.0122m_0$. m_0 is
the free electron mass. Based on the finite element method, we then solved the
Schrödinger equation on a $2 \times 10^3 \text{ \AA}$ circle plane, with the effective mass of the
excitonic quasi particle and aforementioned Coulomb potential. The eigenvalues and
selected eigenfunctions calculated are presented in *Supplementary Figure 18A*. The
exciton transition energy between 1s and 2p is $E_{1s-2p} = 0.275 \text{ eV}$, and the means radius
of wave functions for 1s CT is $\langle \rho_{CT1s} \rangle = 45.7 \text{ \AA}$.

When electron is in graphene and hole is in MoSe₂, the hole and electron
effective masses should be the effective masses of the monolayer MoSe₂ valence band
0.583 m_0 and Al-doped graphene conduction band 0.0135 m_0 , respectively. The
eigenvalues and selected eigenfunctions are presented in Supplementary Figure 18b.
The exciton transition energy between 1s and 2p is $E_{1s-2p} = 0.292$ eV, and the means
radius of wave functions for 1s CT is $\langle \rho_{CT_{1s}} \rangle = 42.3 \text{ \AA}$.

**Supplementary Note 13: Both electron and hole effective masses of graphene are**
**0.012 m_0**

Using 0.012 m_0 reported for a CVD-grown graphene (JAP, 113, 043708, 2013),
we also calculate the interlayer binding energy.

In summary, the binding energy is only slightly affected by the nature of dopant.
In three different calculations, the binding energy is between 0.265~0.292 eV. All are
very close to the experimental value.

**Supplementary Note 14: Origin of carrier reduced mass in graphene and large**
**interlayer binding energy**

It is well known that the effective mass of the pristine graphene is zero, i.e., the
effect mass of the state at $\mathbf{k} = 0$ (Dirac point, at which the Fermi level lie in the
pristine graphene) is zero. However, when the \mathbf{k} point deviates from the Dirac point,
e.g., the Fermi level shifts due to the effect of doping, the effective mass is no longer
zero. We can approximately evaluate this effect based on a tight-binding (TB) model.

Under a TB approach, the energy dispersion of graphene near the Fermi points is
 expressed as Dirac cones:

$$644 \quad E(\mathbf{k}) = \frac{3}{2} t_0 a_0 |\mathbf{k}|, \quad (22)$$

where $a_0 = 2.42 \text{ \AA}$ is the equilibrium C-C bond length, and $t_0 \approx 2.7 \text{ eV}$ is the hopping
 energy between the nearest C atoms. The effective mass at \mathbf{k} is defined as

$$647 \quad m^*(\mathbf{k}) = \frac{\hbar^2}{\frac{\partial^2 E(\mathbf{k})}{\partial \mathbf{k}^2}}. \quad (23)$$

When $\mathbf{k} \neq 0$, the energy dispersion along the normal direction has a finite curvature
 (effective mass) as shown as the solid lines in Supplementary Figure 21. Based on
 Supplementary Equation (22) and (23), the effective mass of electrons at the Fermi
 level (chemical potential μ) along the normal direction is given as

$$652 \quad m^* \Big|_{E(\mathbf{k})=\mu} = \frac{4\hbar^2}{9t_0^2 a_0^2} |\mu|. \quad (24)$$

The dependence of the effective mass on the chemical potential is plotted in
 Supplementary Figure 22. It can be seen that $m^* = 0$ for pristine graphene ($\mu = 0$), but
 it linearly increases with increasing $|\mu|$. For $\mu = -0.17 \text{ eV}$, $m^* \approx 0.039m_0$. If we
 roughly regard the effective mass along the radical direction as unchanged (equal to 0
 as that in pristine graphene), a direct average on both the normal and radical
 directions give $m^* \approx 0.02m_0$ under $\mu = -0.17 \text{ eV}$. This estimated value is actually very
 close to $m^* \approx 0.0116m_0$ from the ab initio calculations assuming B-doped.

In 3D systems, the electric potential of a point charge under the screening of a
 dielectric surrounding with a relative dielectric constant ϵ is described by a screened
 Coulomb potential (in atomic units)

$$V_{3D}(\mathbf{r}) = \frac{1}{\epsilon r} \quad (25)$$

by solving the Poisson's equation, which gives the exciton binding energy of the
hydrogen-like model:

$$E_{b,3D}(\mathbf{r}) = \frac{m^*}{2\epsilon^2}, \quad (26)$$

where m^* is the reduced effective mass. Supplementary Equation (26) suggests that E_b
is proportional to m^* .

However, for a 2D dielectric sheet embedded into vacuum such as single-layer
MoSe₂ or other 2D systems considered in experiments, the 2D screened potential is
not given as a screened Coulomb potential:

$$V_{2D}(\mathbf{r}) \neq \frac{1}{\epsilon r}, \quad (27)$$

but is given as a complicated function in terms of the Struve function and the second
kind Bessel function [refer to: P. Cudazzo, I. V. Tokatly, and A. Rubio, *Phys. Rev. B* **84**,
085406 (2011)], which was also listed as Supplementary Equation (19). As a result,
the binding energy of 2D exciton is not described by Supplementary Equation (26),
but is given as [refer to: T. Olsen, S. Latini, F. Rasmussen, and K. S. Thygesen, *Phys.*
*Rev. Lett.* **116**, 056401 (2016)]

$$E_{b,2D} = \frac{8m^*}{\left(1 + \sqrt{1 + \frac{32\pi\alpha m^*}{3}}\right)^2}, \quad (28)$$

where α is the 2D polarizability calculated as $\alpha = L \frac{\epsilon - 1}{4\pi}$ where ϵ is the dielectric
constant of bulks and L is the thickness of the 2D sheet. A remarkable property in
Supplementary Equation (28) is that E_b is no longer linearly proportional to m^* . When

am^* is large enough, we have $E_{b,2D} \approx \frac{3}{4\pi\alpha}$ which is independent on m^* .
Supplementary Equation 26 is applicable to both undoped and doped 2D systems. An
exciton is formed between an electron and a hole no matter whether the system is
doped or undoped, and the influence of doping just lies in its screening effect. As the
Fermi energy level of boron-doped graphene reaches the experimental value (-0.17
688 eV), the primitive cell size is 26.1 (refer to page 17 of SI), and the doping ratio is
689 actually small (less than 0.2%). Our ab initio calculation shows that the in-plane
dielectric constant of B-doped graphene is almost equal to the value of pristine
graphene when the primitive cell size is larger than 8 (refer to page 19 of SI).
Therefore, the screening effect in doped graphene is very weak.

Take the graphene-MoSe₂ as an example, $\bar{\epsilon} \approx 4.97$ and $L \approx 10.4 \text{ \AA}$, so we have α
$= 6.17$ (in atomic units). E_b is plotted in Supplementary Figure 23 as a function of m^*
based on Supplementary Equation (28), measured with respect to that of $m^* = 0.6$
(the value for MoSe₂). The result of Supplementary Equation (26) is also plotted to
provide a comparison. It can be seen that the dependence of E_b on m^* is highly
nonlinear for the 2D result in Supplementary Equation (28). For $m^* = 0.01m_0$, the 3D
result in Supplementary Equation (26) gives $E_b(m^* = 0.01)/E_b(m^* = 0.6) = 1/60$, i.e.,
the binding energy for $m^* = 0.01$ of graphene-MoSe₂ is 60 times smaller compared to
that of MoSe₂ as indicated by the reviewer, but the 2D result in Supplementary
Equation (28) gives $E_b(m^*=0.01)/E_b(m^*=0.6) \approx 1/3.1$. In other words, with the 2D
equation, the binding energy for $m^* = 0.01$ of graphene-MoSe₂ is merely 3.1 times
smaller compared to that of MoSe₂. Taking the 0.4-0.6 eV binding energy for the latter,

one may expect a binding energy of 0.13-0.19 eV for the graphene-MoSe₂ exciton,
which is close to the results from our measurements. In our work, we used a more
sophisticated theory that has been tested with other 2D systems (JACS 2015, 137,
8313-8320; Nat. Commun. 2016, 7, 12512) to calculate the binding energy and it
turns out to be very close to the experimental value. The dependence of calculated
1s-2p transition energy on reduced mass is plotted in Supplementary Figure 12.

Supplementary References

- 1. Dawlaty JM, *et al.* Measurement of the optical absorption spectra of epitaxial
graphene from terahertz to visible. *Applied Physics Letters* **93**, (2008).
- 2. Gusynin VP, Sharapov SG, Carbotte JP. Unusual microwave response of Dirac
quasiparticles in graphene. *Phys Rev Lett* **96**, (2006).
- 3. Mikhailov SA, Ziegler K. New electromagnetic mode in graphene. *Phys Rev*
*Lett* **99**, (2007).
- 4. Malard LM, Mak KF, Neto AHC, Peres NMR, Heinz TF. Observation of intra-
and inter-band transitions in the transient optical response of graphene. *New J*
*Phys* **15**, (2013).
- 5. Ma Q, *et al.* Tuning ultrafast electron thermalization pathways in a van der
Waals heterostructure. *Nature Physics* **12**, 455-+ (2016).
- 6. Kampfrath T, Perfetti L, Schapper F, Frischkorn C, Wolf M. Strongly coupled
optical phonons in the ultrafast dynamics of the electronic energy and current
relaxation in graphite. *Phys Rev Lett* **95**, (2005).
- 7. Piscanec S, Lazzeri M, Mauri F, Ferrari AC, Robertson J. Kohn anomalies and
electron-phonon interactions in graphite. *Phys Rev Lett* **93**, (2004).
- 8. Li ZZ, Wang JY, Liu ZR. Intrinsic carrier mobility of Dirac cones: The
limitations of deformation potential theory. *J Chem Phys* **141**, (2014).
- 9. Park CH, *et al.* Electron-Phonon Interactions and the Intrinsic Electrical
Resistivity of Graphene. *Nano Lett* **14**, 1113-1119 (2014).
- 10. Lui CH, Mak KF, Shan J, Heinz TF. Ultrafast Photoluminescence from
Graphene. *Phys Rev Lett* **105**, (2010).
- 11. Zhang WJ, *et al.* Ultrahigh-Gain Photodetectors Based on Atomically Thin
Graphene-MoS₂ Heterostructures. *Scientific Reports* **4**, (2014).
- 12. Ugeda MM, *et al.* Giant bandgap renormalization and excitonic effects in a
monolayer transition metal dichalcogenide semiconductor. *Nat Mater* **13**,
1091-1095 (2014).
- 13. Berkelbach TC, Hybertsen MS, Reichman DR. Theory of neutral and charged
excitons in monolayer transition metal dichalcogenides. *Phys Rev B* **88**,
(2013).

14. Hong XP, *et al.* Ultrafast charge transfer in atomically thin MoS₂/WS₂
heterostructures. *Nat Nanotechnol* **9**, 682-686 (2014).
15. Chen HL, *et al.* Ultrafast formation of interlayer hot excitons in atomically
thin MoS₂/WS₂ heterostructures. *Nat Commun* **7**, 12512 (2016).
16. Wang HN, Zhang CJ, Rana F. Ultrafast Dynamics of Defect-Assisted Electron
Hole Recombination in Mono layer MoS₂. *Nano Lett* **15**, 339-345 (2015).
17. Poellmann C, *et al.* Resonant internal quantum transitions and femtosecond
radiative decay of excitons in monolayer WSe₂. *Nat Mater* **14**, 889-+ (2015).
18. Kaindl RA, Hagele D, Carnahan MA, Chemla DS. Transient terahertz
spectroscopy of excitons and unbound carriers in quasi-two-dimensional
electron-hole gases. *Phys Rev B* **79**, (2009).
19. Kresse G, Furthmuller J. Efficient iterative schemes for ab initio total-energy
calculations using a plane-wave basis set. *Phys Rev B* **54**, 11169-11186 (1996).
20. Perdew JP, Burke K, Ernzerhof M. Generalized gradient approximation made
simple. *Phys Rev Lett* **77**, 3865-3868 (1996).
21. Monkhorst HJ, Pack JD. SPECIAL POINTS FOR BRILLOUIN-ZONE
INTEGRATIONS. *Phys Rev B* **13**, 5188-5192 (1976).
22. Grimme S, Antony J, Ehrlich S, Krieg H. A consistent and accurate ab initio
parametrization of density functional dispersion correction (DFT-D) for the 94
elements H-Pu. *J Chem Phys* **132**, (2010).
23. Ramasubramaniam A. Large excitonic effects in monolayers of molybdenum
and tungsten dichalcogenides. *Phys Rev B* **86**, (2012).
24. Zhu XY, Monahan NR, Gong ZZ, Zhu HM, Williams KW, Nelson CA. Charge
Transfer Excitons at van der Waals Interfaces. *J Am Chem Soc* **137**, 8313-8320
(2015).
25. Qiu DY, da Jornada FH, Louie SG. Optical Spectrum of MoS₂: Many-Body
Effects and Diversity of Exciton States. *Phys Rev Lett* **111**, (2013).
26. Tonndorf P, *et al.* Photoluminescence emission and Raman response of
monolayer MoS₂, MoSe₂, and WSe₂. *Opt Express* **21**, 4908-4916 (2013).